# Development of a Submerged Aquatic Vegetation Growth Model in a Coupled Wave-Current-Sediment Model (COAWST v3.4)

5 Tarandeep S. Kalra[1], Neil K. Ganju[2], and Jeremy M. Testa[3]

[1]Integrated Statistics, contracted to the U.S. Geological Survey, Woods Hole, MA 02543, USA
[2]U.S. Geological Survey, Woods Hole, MA 02543, USA
[3] University of Maryland Center for Environmental Science, Solomons, MD, 20688, USA

10 *Correspondence to*: Tarandeep S. Kalra (tkalra@contractor.usgs.gov)

## 0 Abstract

The coupled biophysical interactions between submerged aquatic vegetation (SAV), hydrodynamics (currents and waves), sediment dynamics, and nutrient cycling have long been of interest in estuarine environments. Recent observational studies have addressed feedbacks between SAV meadows and their role in modifying current velocity, sedimentation, and nutrient cycling. To represent these dynamic processes in a numerical model, the presence of SAV and its effect on hydrodynamics (currents and waves) and sediment dynamics was incorporated into the open source model COAWST. In this study, we extend the COAWST modelling framework to account for dynamic changes of SAV and associated epiphyte biomass. Modelled SAV biomass is represented as a function of temperature, light, and nutrient availability. The modelled SAV community exchanges nutrients, detritus, dissolved inorganic carbon, and dissolved oxygen with the water-column biogeochemistry model. The dynamic simulation of SAV biomass allows the plants to both respond to and cause changes in water column and sediment bed properties, hydrodynamics, and sediment transport (i.e., a two-way feedback). We demonstrate the behavior of these modelled processes through application to an idealized domain, then apply the model to a eutrophic harbour where SAV dieback is a result of anthropogenic nitrate loading and eutrophication. These cases demonstrate an advance in the deterministic modelling of coupled bio-physical processes and will further our understanding of future ecosystem change.

## 1 Introduction

Submerged aquatic vegetation (SAV), or seagrasses, are rooted vascular plants that inhabit sediments of estuaries and coastal waters, with a wide global distribution. SAV are important primary producers in shallow environments, provide habitat for a number of aquatic organisms, can slow water velocities and dampen wave energy to trap particulate material (Carr et al., 2004), and can alter biogeochemical cycles through oxygenation of sediments (Larkum et al., 2006). The positive impact of ecosystem services provided by SAV presence has been well-studied (Hemminga and Duarte, 2000, Nixon et al., 2001, Terrados and Borum, 2004. and McGlathery et al., 2007, Hayn et al., 2014). The growth of SAV is dependent upon light availability at the leaf surface, which is a function of light attenuation in the water-column and the biomass of epiphytic algae growing on SAV stems. During the last several

decades, the loss of SAV has accelerated owing to anthropogenic pressures (Kennish et al., 2016) or natural causes such as storms (Hamberg et al., 2017). One of the dominant factors of SAV loss is eutrophication through nutrient loading, exemplified by increased phytoplankton growth and epiphytic growth on vegetation. This results in a reduction of light availability (Burkholder et al., 2007), causing a loss of SAV habitat (Cabello-Pasini et al., 2003, Short and Neckles, 1999).

     The complex interactions between light availability, nutrient loading, SAV dynamics, hydrodynamics, and sediment transport can be investigated using numerical modelling tools. Few modelling efforts have attempted to couple the effects of hydrodynamics and light availability to model the growth of SAV. Everett et al., 2007 and Hipsey and Hamilton, 2008 coupled the effects of chlorophyll and water to account for SAV variability while Bissett et al., 1999a, 1999b used spectral underwater irradiance to model the light availability required for SAV growth. Carr et al., 2012a, 2012b developed a one-dimensional coupled hydrodynamics, sediment, and vegetation growth dynamics model. The model solved for vertical 1-D dynamics of SAV growth through a change in biomass that depended on water temperature, irradiance and seagrass properties Ganju et al., 2012 used a three-dimensional circulation model (ROMS) coupled to a Nutrient Phytoplankton Zooplankton Detritus (NPZD) eutrophication (water column bio-geochemistry model) developed by Fennel et al., 2006 and integrated the spectral light attenuation formulation (bio-optical model) provided by Gallegos et al., 2011. These models were linked to a benthic seagrass model to calculate seagrass distribution (Zimmerman et al., 2003) and applied on the temperate estuary of West Falmouth Harbor (del Barrio et al., 2014). While the model was able to capture the loss of SAV due to insufficient light, it did not include interactions with epiphytes or exchanges with water-column nutrient and gas pools. The hydrodynamic feedbacks (change in currents and waves) and morphodynamic changes (sediment distribution) due to presence of SAV were also ignored. While these dynamic processes have significant implications for coastal ecosystem resilience, numerical models that allow for the two-way feedbacks between hydrodynamics, sediment transport, and SAV growth and nutrient cycling have generally been lacking.

     Recently, Beudin et al. 2017 implemented the physical effects of SAV in a vertically varying water column through momentum extraction, vertical mixing as well as accounting for wave dissipation due to vegetation. These processes were implemented within the open source COAWST (Coupled-Ocean-Atmospheric-Wave-Sediment Transport) modelling system that couples the hydrodynamic model (ROMS), the wave model (SWAN) and the Community Sediment Transport Modelling System (CSTMS) (Warner et al., 2010). Through this effort, the COAWST framework accounted for the coupled seagrass-hydrodynamics interactions. The model reproduced the turbulent shear stress across the canopy interface and peaked at the top of the canopy similar to the observations of Ghisalberti and Nepf (2004, 2006). The presence of seagrass patch led to a reduced shear-scale turbulence within the canopy and an enhanced wake-scale generated turbulence. For more details on the impact of seagrass on hydrodynamics, readers are referred to Beudin et al. 2017. The inclusion of the physical effects of SAV on flow and sediment dynamics (Beudin et al., 2017) in COAWST allows us to develop a framework that results in dynamic growth of SAV using the temperature, nutrient loading and light availability in the water column. Therefore, in this work we implement a SAV growth model that dynamically changes the SAV properties (stem density and height). The growth of SAV is modeled

as biomass which includes the above ground (stems and shoots), below ground (roots and rhizomes) biomass and epiphyte biomass. Individual biomass equations described in the implementation of SAV growth model (section 2.2) are based upon previous SAV biomass models (primarily Madden and Kemp 1996), some of which have been previously implemented to simulate growth conditions for SAV in three-dimensional numerical model simulations (e.g. Cerco and Moore 2001). The change in biomass leads to a spatial and temporal variation of SAV density and height. With the inclusion of the SAV growth model, SAV can grow or dieback while contributing and sequestering nutrients from the water column (modifying the biological environment), and subsequently affect the hydrodynamics and sediment transport (modifying the physical environment). Conversely, a change in the physical environment, for instance the amount of sediment in the water column, can decrease light availability, and cause SAV dieback leading to reduced wave attenuation, increased sediment resuspension, and a further decrease of light availability.

We demonstrate the two-way biophysical coupling framework as follows: the SAV growth model and integration into COAWST are discussed in section 2; in section 3, the model setup for the idealized domain and a realistic simulation of West Falmouth Harbor, MA are described; in section 4, we present the results from the two model configurations along with a discussion of limitations of the current modelling work and in section 5, we summarize our work and outline areas of future research.

## 1    Methods

### 2.1 Inclusion of SAV effect on flow and sediment dynamics in the numerical model

Beudin et al. (2017) implemented the parameterizations that accounted for the presence of SAV within a coupled hydrodynamic and wave model within the open-source COAWST numerical modelling system (Warner et al., 2008). The COAWST framework utilizes ROMS (Regional Ocean Modelling System) for hydrodynamics with a wave model - SWAN (Simulating WAves Nearshore) via the Model Coupling Toolkit (MCT) generating a single executable program (Warner et al., 2008). ROMS (Regional Ocean Modelling System) is a three-dimensional, free surface, finite-difference, terrain-following model that solves the Reynolds-Averaged Navier-Stokes equations using the hydrostatic and Boussinesq assumptions (Haidvogel et al., 2008). The transport of turbulent kinetic energy and generic length scale are computed with a generic (GLS) two-equation turbulence model. SWAN (Simulating WAves Nearshore) is a third-generation spectral wave model based on the action balance equation (Booij et al., 1999). In ROMS, the presence of SAV extracts momentum, adds wave-induced streaming, and generates turbulence dissipation. Similarly, the wave dissipation due to vegetation modifies the source term of the action balance equation in SWAN. All these sub-grid scale parameterizations account for changes due to vegetation in the water column extending from the bottom layer to the height of the vegetation. SWAN only accounts for wave dissipation due to vegetation at the bottom layer. The coupling between the two models occurs with an exchange of water level and depth averaged velocities from ROMS to SWAN and wave fields from SWAN to ROMS after a desired number of time steps. The vegetation properties are separately input in the two models at the beginning of the simulations. Through these changes, the SAV can affect the bottom stress calculations that determine the resuspension and transport of sediment, providing a feedback loop between SAV-sediment dynamics-hydrodynamics and wave dynamics. To account for sediment dynamics, the Community Sediment Transport Modelling System (CSTMS) (Warner et al., 2010) is used to

track the transport of suspended-sediment and bed load transport under the action of current and wave-current forcing. The model can represent an unlimited number of user defined sediment classes.

## 2.2 SAV growth model

The SAV growth model is primarily based upon a previous growth model developed and implemented in Chesapeake Bay by Madden and Kemp (1996). The model simulates the temporal dynamics of above ground biomass (AGB) that consists of stems or shoots, and the below ground biomass (BGB) that consists of roots or rhizomes. In addition to AGB and BGB, epiphytic algal biomass (EPB) is simulated to account for reductions in light availability to plant leaves due to shading of SAV leaves by epiphytes under high nutrient loading conditions. AGB, BGB and

EPB are simulated as total biomass per unit area, with nitrogen as the currency for biomass. Changes in AGB and BGB pools are simulated as a function of primary production and respiration, mortality (e.g. grazing), and nitrogen exchange through the seasonal translocation of nitrogen between roots and shoots. EPB are modelled as a function of primary production, respiration, and mortality.

The remaining section describes the source terms that calculate the evolution of AGB, BGB and EPB and denoted by

$\alpha$, $\beta$ and $\gamma$ respectively. Table 1 and Table 2 describe model variables and input parameters along with their equivalent symbols used in the source code.

2.1 Primary production ($\rho_s$): The primary production of $\alpha$ depends on the maximum potential growth rate ($\mu_s$) and downward deviations from this maximal rate resulting from light ($\varphi_s$) and nutrient ($\vartheta_s$) availability as:

$$\rho_s = \mu_s \min(\varphi_s, \vartheta_s) \tag{1}$$

The maximum potential growth ($\mu_s$) can be described as:

$$\mu_s = \lambda_s \, \vartheta_s \, g_s \exp[r_s \left(\frac{1.0}{T-T_o}\right)] \tag{2}$$

where $\lambda_s$ is a self-shading parameter that accounts for crowding and self-shading within the SAV canopy, $g_s$ accounts for SAV's maximum growth fraction, $r_s$ is the active SAV respiration coefficient, $T$ is the temperature in water column, $T_o$ is the user defined optimum temperature that allows for species specific sensitivities to temperature. The

self-shading parameter, $\lambda_s$ used in Eq. 3 is calculated by setting a maximum aerial biomass of SAV (Madden and Kemp 1996), thereby making growth rates density-dependent and is defined as:

$$\lambda_s = 1 - \left(\frac{\alpha}{\lambda_{s,max}}\right)^2 \tag{3}$$

where $\alpha$ is the above ground SAV biomass and $\lambda_{s,max}$ accounts for the maximal SAV biomass.

The availability of photosynthetically active radiation (PAR) represented by mathematical symbol $\theta$ for SAV leaves in the bottom cell is simulated using a bio-optical model (Gallegos et al. 2009, del Barrio et al. 2014). While the bio-optical model generates predictions of light available across the spectrum within PAR, the light availability ($\varphi_s$) used to compute primary production (Eq. 1) is obtained through traditional photosynthesis-irradiance (PI) curves based on total PAR used to represent SAV growth responses to light:

$$\varphi_s = \frac{\theta}{l_s + \theta} \tag{4}$$

where $l_s$ is the half-saturation for light limitation for SAV and $\theta$ refers to photosynthetically available radiation that is obtained from the bio-optical model. This simplified PI formulation, which has been applied in previous SAV models (Madden and Kemp 1996, Zaldívar et al. 2009, Jarvis et al. 2014) is applied so that a general and flexible SAV growth response is available for users in a wide-variety of environments with different species. More complex approaches can easily be applied (e.g. Zharova et al. 2001, Carr et al. 2012). The nutrient limitation ($\vartheta_s$) required in Eq.1 to compute primary production represents the fact that rooted plants can obtain nutrients from both sediments (as in Madden and Kemp, 1996) and the water-column and is defined as:

$$\vartheta_s = DIN_{wc} + \frac{n_{s,1} DIN_{sed}}{n_{s,2} DIN_{sed} + n_{s,1} DIN_{sed}} \tag{5}$$

where $DIN_{wc}$ is the dissolved inorganic nitrogen concentration in the water column based on the sum of $NH_4$ (Ammonium) and $NO_3$ (Nitrate) in the water column and $DIN_{sed}$ is the amount of dissolved inorganic nitrogen ($DIN = NH_4 + NO_3$) in the sediment bed layer, and $n_{s,1}$ is the half-saturation for nutrient limitation for SAV roots.

2.2.2 Respiration: SAV respiration terms are partitioned into active and basal respiration, where the active respiration term represents respiration that is dependent on the photosynthesis rate, and the basal rate represents maintenance respiration rate. The active respiration term is defined as:

$$\delta_s = \rho_s \, p_s \exp(r_s \, T) \tag{6}$$

where $\rho_s$ is the primary production term (Eq. 1), $p_s$ is the maximum fraction of photosynthesis available for respiration, $r_s$ is the SAV's active respiration coefficient, $T$ is the temperature in the water column. The above ground basal respiration term is defined as:

$$\varepsilon_s = c_s \exp(h_s \, T) \tag{7}$$

where $c_s$ is the maximum fraction of SAV below ground biomass (BGB) that is respired, $h_s$ is the SAV basal respiration coefficient for both AGB and BGB, $T$ is the temperature in the water column.

2.2.3 Mortality: The mortality of SAV is computed separately for above-ground and below-ground biomass, where above ground mortality ($\omega_{s,1}$) accounts for the sloughing of leaves and grazing in combination as:

$$\omega_{s,1} = (m_\alpha \, \alpha)^2 \tag{8}$$

where $m_\alpha$ is the above ground SAV mortality rate (sloughing).

Below ground mortality, $\omega_{s,2}$, is a function of temperature and is given as:

$$\omega_{s,2} = 0.01 \, \beta \exp(m_\beta \, T) \tag{9}$$

where $m_\beta$ is the below-ground SAV mortality rate. Additional terms include that modify the AGB and BGB include the seasonal exchange (translocation) of root material (nitrogen) quantified as a fraction of primary production and the translocation of BGB to AGB which represents the seasonal translocation of nitrogen from roots to stems as the plants initially emerge in spring. Each of these terms is initiated on a specified day of the year (Madden and Kemp 1996), and can be altered to account for species differences or regional differences in the physiology of particular species. Translocation of nitrogen from leaves to roots/rhizomes (storage) is modelled as a continuous response to

SAV primary production ($\rho_s$) and is given by defining $\sigma_{s,1}$ (translocation of above ground biomass to below ground biomass) as:

$$\sigma_{s,1} = \rho_s \, k_1 \tag{10}$$

where $k_1$ is a downward translocation coefficient.

and translocation from roots/rhizomes to leaves (upward translocation) is modelled as a simple linear function of below ground biomass (denoted by $\beta$) that begins after a user-defined threshold temperature is crossed and is given by defining $\sigma_{s,2}$ (translocation of below ground biomass to above ground biomass) as:

$$\sigma_{s,2} = \beta \, k_2 \tag{11}$$

where $k_2$ is a upward translocation coefficient.

The epiphyte biomass (EPB) is computed similarly to SAV biomass by simulating EPB as a function of primary production, respiration, and mortality (e.g. grazing).

2.2.4 Primary production ($\rho_e$): The primary production of EPB depends on the maximum potential growth rate ($\mu_e$) and a limitation between light ($\varphi_e$) and nutrient ($\vartheta_e$) availability, as:

$$\rho_e = \mu_e \, \min(\varphi_e, \vartheta_e) \tag{12}$$

The maximum potential growth of EPB ($\mu_e$) can be described as:

$$\mu_e = \lambda_e \, \vartheta_e \, g_e \, \exp[r_e \left(\frac{1.0}{T - T_{e,o}}\right)] \tag{13}$$

where $\lambda_e$ is the self-shading parameter that accounts for spatial limits on the epiphyte population, $g_e$ accounts for epiphyte's maximum growth fraction, $r_e$ is the active epiphyte respiration coefficient, $T$ is the temperature in water

column, $T_{e,o}$ is the user defined optimum temperature that allows for species-specific sensitivities to temperature. $\lambda_e$ is calculated by setting a maximum aerial biomass of EPB, thereby making growth rates density-dependent similar to the SAV growth rate, as:

$$\lambda_e = 1 - \left(\frac{EPB}{\lambda_{e,max}}\right)^2 \tag{14}$$

where $EPB$ is the epiphyte biomass and $\lambda_{e,max}$ is the maximum epiphyte biomass.

The light availability ($\varphi_e$) used to compute primary production (Eq. 10) is obtained through traditional photosynthesis-irradiance (PI) curves used to represent epiphyte growth response to light, as:

$$\varphi_e = \frac{\theta}{l_e + \theta} \tag{15}$$

where $l_e$ is the half-saturation for light limitation for epiphytes and $\theta$ is the photosynthetically available radiation

obtained from the bio-optical model. The nutrient limitation ($\vartheta_e$) required in Eq.1 to compute primary production for epiphytes depends only on the nutrients in the water-column and is a traditional algal form (e.g. Monod model) given as:

$$\vartheta_e = \frac{n_e \, DIN_{wc}}{n_e \, DIN_{wc} + n_e} \tag{16}$$

where $DIN_{wc}$ is the amount of dissolved inorganic nitrogen in the water column, $n_e$ is the half-saturation for nutrient limitation for epiphytes.

2.2.5 Respiration: Epiphyte respiration terms are partitioned into active and basal respiration, where the active respiration term represents respiration that is dependent on the photosynthesis rate, the basal rate represents the maintenance respiration rate. The active respiration term is defined as:

$$\varepsilon_{e,1} = \rho_e\, p_e \exp(r_e\, T) \tag{17}$$

where $\rho_e$ is the primary production term (Eq. 1), $p_e$ is the maximum fraction of photosynthesis for epiphytes, $r_e$ is the epiphyte's active respiration coefficient, $T$ is the temperature in the water column.

The basal respiration term is defined as:

$$\varepsilon_{e,2} = c_e \exp(h_e\, T) \tag{18}$$

2.2.6 Mortality: The mortality of epiphytes depends on mortality and grazing of algal cells, as well as losses associated with SAV sloughing (which effectively removes epiphytes from a cell). The mortality term is given as a simple linear form:

$$\omega_e = m_e\, \gamma \tag{19}$$

where $m_e$ is the epiphyte mortality rate. The loss of epiphyte biomass due to grazing ($\tau_{e,1}$) modelled using an Ivlev function can be described as:

$$\tau_{e,1} = z_{e,max}\,[1.0 - \exp(-z_e)] \tag{20}$$

where $z_{e,max}$ is the maximum grazing rate on epiphytes and $z_e$ is the grazing coefficient on epiphytes. The reduction of epiphyte biomass due to the SAV sloughing loss is computed as:

$$\tau_{e,2} = \left(\frac{\omega_{s,1}\, t}{\alpha}\right) \tag{21}$$

where $\omega_{s,1}$ is the above ground mortality term described in Eq. 8, is the time step size in per day units and $\alpha$ refers to the above ground biomass.

The above ground biomass (AGB) computed in the SAV growth model is utilized to obtain SAV shoot height (meters) and stem density (stems/m$^2$), to allow for the biomass model (AGB) to be translated into variables input into the SAV-hydrodynamic coupling. The shoot height ($l_v$) is related to AGB (denoted by $\alpha$) as:

$$l_v = 0.45 \left(\frac{\alpha}{120.0 + \alpha}\right) \tag{22}$$

The relationship is based on measurements of *Zostera marina* in Chincoteague Bay and Chesapeake Bay (Fig. 2), but is consistent with relationships for *Z. marina* determined elsewhere (Krause-Jensen et al., 2000). Other three-dimensional models have used similar formulations (e.g. Cerco and Moore, 2001 for Chesapeake Bay). SAV stem density $n_v$, (in stems/m$^2$) is computed from a similar empirical formulation based on relationships in Krause-Jensen et al., 2000 and is computed as:

$$n_v = 4.45\, \alpha \tag{23}$$

**2.3 Integration of SAV growth model with Water-Column Biogeochemistry Model (BGCM model)**

The SAV growth model is built to interact dynamically with the water-column biogeochemistry model (BGCM model) within the COAWST modelling framework. We utilize one of the existing BGCM models developed by Fennel et al., 2006 that accounts for nutrients ($NO_3$, $NH_4$), phytoplankton and zooplankton biomass, and detritus. The BGCM model

in the current simulations solved for twelve state variables. The spectral irradiance model that provides the light attenuation in response to chlorophyll, sediment, and CDOM was previously integrated (Gallegos et al. 2009, del Barrio et al. 2014) into the BGCM model. The BGCM model was implemented within the hydrodynamic component of COAWST model, ROMS (Regional Ocean Modeling System). ROMS is a three-dimensional, free surface, terrain-following numerical model that solves finite-difference approximations of the RANS equations using the hydrostatic

and Boussinesq assumptions (Chassignet et al., 2000 and Haidvogel et al., 2000). ROMS is discretized in horizontal dimensions with curvilinear orthogonal Arakawa C grid (Arakawa, 1966). Each state variable is calculated based on the tracer transport equation with tracer concentrations calculated at the grid cell centers as follows:

$$\frac{\partial c}{\partial t} + u\frac{\partial c}{\partial x} + v\frac{\partial c}{\partial y} + w_d\frac{\partial c}{\partial z} = \frac{\partial}{\partial z}\left(v\frac{\partial c}{\partial z}\right) + C_{source} \tag{24}$$

where $C$ is the tracer quantity, $t$ is time, $x$ and $y$ are the horizontal coordinates and $z$ is the vertical coordinates. $u$ and $v$ are the horizontal components of current velocity with $w_d$ being the sinking velocity for tracers such as detritus. $v$ is the turbulent diffusivity coefficient and $C_{source}$ is the tracer source/sink term, which represents the net effects of all sources and sinks in this representation. There are several choices of advection schemes for tracer advection available

in COAWST (Kalra et al., 2019) and in the current simulations, we utilized Multidimensional Positive Definite Advection Transport Algorithm (MPDATA) scheme (Smolarkiewicz, 1984) that has been derived from Lax Wendroff (LW) family of schemes. The time marching scheme for tracers involves a predictor-corrector step using the leapfrog-trapezoidal methods. The 3-D tracer equations are solved at a different and shorter time step than the depth-integrated 2-D barotropic equations. The integration between the baroclinic mode and barotropic mode is performed using a

split-explicit time step approach (Shchepetkin and McWilliams, 2005, 2009). The predictor step calculates the tracer values that updates the momentum equations at an intermediate time step. At that point, the split-explicit algorithm is executed and the update of tracers is done using the corrector step after the new values of velocity are available. For more details of this algorithm, readers are readers are referred to Shchepetkin and McWilliams, 2005 and 2009. The vertical tracer diffusion terms are solved using a fourth-order centered scheme (Shchepetkin and McWilliams, 2005).

The vertical advective fluxes are computed using the piecewise parabolic method (Colella and Woodward, 1984). The vertical terms utilize a backwards Euler method for time marching.

The changes in water-column variables (dissolved and particulate nitrogen, dissolved oxygen, dissolved inorganic carbon) due to the SAV growth model occur locally at the bottom cell through the source terms ($C_{source}$) that affect six state variables in the BGCM model: $NO_3$ (Nitrate), $NH_4$ (Ammonium), $DO$ (Dissolved Oxygen), $CO_2$

(Carbon dioxide), $LDeN$ (Labile Detrital Nitrogen), $LDeC$ (Labile Detrital Carbon). The change in these state variables based on the SAV growth model is as follows:

$$\frac{\partial DIN_{SAV}}{\partial t} = (\delta_s + \varepsilon_s - \rho_s)(1 - s_f)t + (\varepsilon_{e,1} + \varepsilon_{e,2} - \rho_e)t \tag{25}$$

where $\frac{\partial DIN_{SAV}}{\partial t}$ is the net impact of SAV and epiphyte growth on water-column nitrogen concentrations and $s_f$ decides

the portioning of nutrient uptake between sediment and water column using a logistic function and is defined as:

$$s_f = 1 - \left(\frac{1}{1+\exp[-m_f(DIN_{wc}-k_f)]}\right) \tag{26}$$

where $m_f$ and $k_f$ are constants and equal to 0.2 and 15.0 respectively and $DIN_{wc}$ (Dissolved Inorganic Nitrogen) is calculated as a sum of state variables $NH_4$ (Ammonium) and $NO_3$ (Nitrate) in the water column. If net growth from SAV and epiphytes is negative, the net nitrogen regeneration is realized as $NH_4$ production in the water column

$(\frac{\partial NH_4}{\partial t} = \frac{\partial DIN_{SAV}}{\partial t})$. If there is net growth originating from SAV and epiphytes, the associated water column uptake of DIN is apportioned between $NO_3$ and $NH_4$ relative to their availability in the water-column via the following equations:

$$\frac{\partial NH_4}{\partial t} = \left(\frac{\partial DIN_{SAV}}{\partial t}\right)\left(\frac{NH_4}{DIN_{wc}}\right) \tag{27}$$

$$\frac{\partial NO_3}{\partial t} = \left(\frac{\partial DIN_{SAV}}{\partial t}\right)\left(\frac{NO_3}{DIN_{wc}}\right) \tag{28}$$

$$\frac{\partial DO}{\partial t} = \left(\rho_s - \delta_s - \varepsilon_s + \rho_e - \varepsilon_{e,1} - \varepsilon_{e,2}\right)t \tag{29}$$

$$\frac{\partial CO_2}{\partial t} = \left(\delta_s + \varepsilon_s - \rho_s + \varepsilon_{e,1} + \varepsilon_{e,2} - \rho_e\right)t \tag{30}$$

$$\frac{\partial LDeN}{\partial t} = \left(\omega_{s,1} + \omega_e + \tau_{e,1}\right)t \tag{31}$$

$$\frac{\partial LDeC}{\partial t} = \left(\omega_{s,1} + \omega_e + \tau_{e,1}\right)t \tag{32}$$

All the source terms in equations (25 and 27-32) are solved using the SAV growth model described in Section 2.2 and

in equation 30 and 32, these terms are converted to moles of Carbon from moles of Nitrogen assuming a fixed (and user-defined based on local data) C:N ratio in SAV tissue (we assumed a C:N of 30).

**2.4 Two-way feedback from SAV to hydrodynamics, waves, sediment dynamics, and biogeochemistry**

The addition of the SAV growth model leads to the biological evolution of SAV properties based on temperature,

light, and nutrient availability. The modelled SAV community exchanges nutrients. detritus, dissolved oxygen, and dissolved inorganic carbon with the water-column BGCM. Changes in SAV biomass, and canopy characteristics also

impacts hydrodynamics, wave dynamics and sedimentary dynamics (resuspension-transport). By lowering the current speed and attenuation of wave flow, the reduction in bed shear stresses in the vegetation canopy reduces sediment resuspension; thereby altering sediment transport in the model (as described in Section 2.1), that feedback to control light availability and, in turn, potential seagrass biomass production. This methodology of including the SAV growth model enables the COAWST framework to have a two-way feedback between hydrodynamic-biological coupling. Figure 1 describes the coupling process between different modules schematically.

## 3. Model Setup

### 3.1 Idealized test case

The implementation of the SAV growth model within the COAWST framework is first tested on an idealized domain. The test case consists of an idealized rectangular domain of 9.2 km width and 9.8 km length with a 1 m deep basin. The number of interior domain points are 90 in the x-direction and 98 in the y-direction with 10 vertical sigma layers. The resulting domain has a grid resolution of 100 m by 100 m in horizontal and 0.1 m in the vertical (this varies with water level). A rectangular vegetation bed extends from the north boundary of the domain southward 8 km, with a width of 1.8 km, centered in the domain (Figure 3). The ROMS barotropic and baroclinic time steps are 0.05 s and 1 s respectively. The bed roughness is set to $z_o$ =1.5 mm. The k − ε turbulence model is implemented following the GLS method (Warner et al., 2005). The initial AGB, BGB and EPB in the vegetation bed are set to be 90, 15 and 0.01 mmol N/m$^2$ respectively. The vegetation density, height, diameter and thickness are initialized to be 400 stems/m$^2$, 0.19 m, 1.0 mm and 0.1 mm respectively. The vegetative drag coefficient (C$_D$) is set to be 1 (typical value for a cylinder at high Reynolds number). The imposed surface wind speed is 3 m/s from the north to induce a wave field. The surface air pressure is initialized as 101.3 kPa. The kinematic surface solar shortwave radiation is set to an amplitude of 500.0 W/m$^2$ with a 24-hour period. The kinematic surface longwave radiation flux is set to zero (W/m$^2$). The surface air temperature varies between 1.5 ℃ to 18.5 ℃ over a yearly period. The surface solar downwelling spectral irradiance just beneath the sea surface is set following Gregg and Carder (1990). The cloud fraction is set to be zero. The bulk flux parameterizations in COAWST for surface wind stress and surface heat flux are based on the COARE code (Fairall et al. (1996a, 1996b) and Liu et al. (1979)).

The model is forced by oscillating the water level on the northern boundary with a tidal amplitude of 0.25 m and a period of 12 hours. Northern boundary conditions include a water temperature variation between 1.5 ℃ to 18.5℃ over an yearly period. Salinity and $NO_3$ at the northern boundary are set to 35 psu and 20 mmol N/m$^3$ respectively, and we impose a suspended sediment concentration of 0.5 g/L as well. The northern boundary condition for tracers is a radiation condition with nudging on a 6h timescale. For both flow and tracer fields (physical and biological), the western and eastern boundaries have a gradient condition and the southern boundary is closed. The model setup for the idealized domain is simulated for 60 days and the model output is averaged over each day.

### 3.2 Realistic test case: West Falmouth Harbor, Massachusetts, USA

del Barrio et al. (2014) used an offline coupling of the COAWST model with a bio-optical seagrass model (Zimmerman et al., 2003) to study the influence of nitrate loading and sea-level rise on seagrass presence/absence in

West Falmouth Harbor, Massachusetts, USA. Nitrate concentrations in groundwater exceeded 200 µM due to a wastewater treatment plant in the watershed, however recent mitigation is expected to eliminate the nitrate load in the future. The model of del Barrio et al. (2014) used the biogeochemical results to generate spectral irradiance fields which were then passed to the bio-optical model. While useful for investigating the interaction between phytoplankton dynamics, light climate, and potential seagrass coverage, that model did not account for the interaction of seagrass with water column and sediment nitrogen pools, or hydrodynamics. Therefore, we tested the fully coupled hydrodynamic, biogeochemical, and vegetation model using the same hydrodynamic and biogeochemical model setup (Ganju et al., 2012 and del Barrio et al., 2014), but with the full vegetative interaction implemented. Briefly, the model is forced with tides at the western boundary, groundwater and nitrate loading at the eastern boundary, and solar irradiance at the air-sea boundary. Further details on the model setup are given by Ganju et al. (2012) and del Barrio et al. (2014). The hydrodynamic and biogeochemical (e.g. chlorophyll concentrations, light attenuation) results were assessed in those studies. In this work, we test the ability of the coupled model to reproduce the present-day spatial pattern of seagrass presence, with growth and persistence expected in the outer harbor, and dieback in the inner harbor, where nitrate loading, phytoplankton growth, and light attenuation are highest. The initial SAV properties include a plant height of 0.195 m, plant density of 110 stems/m$^2$, plant diameter of 0.001 m, and plant thickness of 0.0001 m. The vegetative drag coefficients $C_D$ in the flow model and the wave model are set to 1 (typical value for a cylinder at high Reynolds number). We utilize the SAV growth model parameters described in Table 1. The model setup for West Falmouth Harbor (Section 3.2) is simulated for 56 days, beginning 2 July 2010 (Ganju et al., 2012).

## 4 Results and Discussion

### 4.1 SAV, sediment, and hydrodynamics in the idealized test case

Simulations of the coupled hydrodynamic-biogeochemical-SAV model revealed the integrated nature of estuarine dynamics in response to submerged macrophytes. In these simulations, SSC was imposed at the northern open boundary at concentrations of 0.5 g/L (and zero g/L within the bed), resulting in a decline in SSC as one moves towards the southern boundary (Fig. 4a). This distribution of SSC input results in an increase in light attenuation ($K_{dpar}$=30.0 m$^{-1}$) in the region close to the northern boundary (0.0 km), while background conditions prevail in the southern reaches (Fig. 4b). In Fig. 4b, SSC input from the northern boundary causes a decrease in light availability within the modelled SAV region between the open boundary in the north and about 2.4 km into the SAV bed. Consequently, these sub-optimal light conditions in the northern 2.4 km of the SAV bed cause AGB to decrease from its initial value of 90.0 millimoles N/m$^2$ to 30.0 millimoles millimoles/m$^2$ (Fig. 5a). Boundary effects associated with SSC inputs are substantially muted in the region between 2.4 km and 8.0 km within the SAV bed (Figs. 4&5), where in-bed SSC concentrations are much lower than those outside the bed at the same distance from the boundary. As a consequence, where AGB biomass increases from its initial value of 90.0 millimoles N/m$^2$ to 150.0 millimoles N/m$^2$ over the course of the simulation. Increases in SAV biomass within the bed during the simulation led to increases in SAV density and height, where SAV density increased from its initial value of 400 stems/m$^2$ to of 810 stems/m$^2$ owing to favourable light conditions from y=2.4 km to y=8.0 km. Thus, the model captured the role of SAV in resisting SSC transport into the bed, allowing for greater light availability and an increase in growth rates and biomass accumulation.

The temporal evolution of SAV biomass in response to the SSC input at the northern boundary further emphasizes the self-stimulating role of SAV in the idealized simulations. A comparison of model simulations at two locations within the initially described SAV bed of the idealized domain (indicated in Fig. 5a and corresponding to y=0.1 km and y=4.5 km from the northern boundary) reveal that close to the northern boundary (y=0.1 km), the daily averaged light attenuation remains high (above 30 m$^{-1}$) over the 60-day period (Fig. 5a). At y=0.1 km, the increased light attenuation in the northern location corresponds to the lack of light availability and this causes a decay of AGB from its initial value of 90.0 millimoles N/m$^2$ to 30.0 millimoles N/m$^2$. (Fig. 5b). This decay in AGB over the 60-day period at y=0.1 km (SAV dieback), contrasts sharply with the AGB increases inside the SAV bed at the southern location (y=4.5 km), where light attenuation is lower because sediments have not penetrated the SAV bed, allowing for higher SAV growth rates. The higher SAV growth rate inside the SAV bed at y=4.5 km can be observed (Fig. 5c) by looking at the net primary production of SAV ($\rho_s - \delta_s - \varepsilon_s$). At this location (y=4.5 km), the SAV growth rate increases over the 60-day period while it keeps decreasing in the northern location (y=0.1 km). Due to the higher SAV growth inside the SAV bed (y=4.5 km), the SSC in the bottom cell remains low (Fig. 5d) and at y=0.1 km due to the SAV dieback, the sediment concentration in the water column stays high and above 0.25 g/L.

As mentioned above, the SSC input on the northern boundary of the idealized domain causes a region of sub-optimal light conditions that lead to the SAV dieback; while the SAV growth occurs in the remaining bed where favourable light conditions exist. The effect of change in SAV density and height on the hydrodynamics and morphodynamics at the end of the simulation can be demonstrated by using the same idealized domain. To this end, two transects are chosen that are along the length of the SAV bed and extend from the northern boundary towards the southern boundary. The transects are chosen inside at x=1.8 km (outside of the SAV bed) and at x=4.8 km (inside of the SAV bed). The depth-integrated SSC and bottom stresses averaged on the 60$^{th}$ day in the transect (Fig. 7a) outside of the SAV bed show that the profile of bottom stress follows the distribution of SSC along the transect. In Fig. 7a, a 0.2 N/m$^2$ of peak bottom stress is obtained at x=1.8 km (outside of the SAV bed) that corresponds to a depth-averaged SSC of 0.31 g/L. On the other hand, the transect within the SAV bed (Fig. 7b) shows that the region where SAV dieback has occurred (between 0.0 km to 2.4 km) corresponds to increased bottom stresses (0.13 N/m$^2$ at the north most location and a corresponding SSC of 0.26 g/L) while the region where the SAV growth has occurred, the bottom stresses are close to zero (i.e. from 2.4 km and onwards).

The simulation of the idealized domain demonstrates the capability of the modelling framework to perform two-way feedbacks between hydrodynamics, sediment and biological dynamics. The SSC input in the northern boundary affects the light attenuation in the domain and causes SAV dieback close to the northern boundary. The SAV grows in the region where favourable light conditions exist. The SAV dieback leads to increased bottom stresses while the growth of SAV leads to a decrease in bottom stresses; illustrating the fact that the SAV act as bottom sediment stabilizers by reducing SSC.

## 4.2 SAV growth in West Falmouth Harbor

The present-day simulation of seagrass dynamics reproduces the patterns of chlorophyll (via phytoplankton), light attenuation, and near-bottom PAR simulated by del Barrio et al., 2014. Nitrate loading from shoreline point sources

led to increased phytoplankton growth indicated by increased chlorophyll and light attenuation in the landward, northeast portion of the harbor (Fig. 8a,b), while bathymetric controls in the deeper central basin led to decreased near-bottom PAR (Fig. 8c). Peak AGB exceeds 100 millimoles N m$^{-2}$, while seagrass presence begins towards decline in the inner harbor and in the central basin as expected. Intertidal areas around the periphery of the harbor are devoid of AGB due to the enforced masking of areas with intermittent wetting and drying.

Time-series of these parameters (Fig. 9) from selected outer and inner harbor locations over the first 22 days demonstrate the diurnal variability, as well as the rapid loss of AGB in the inner harbor due to the local nitrate loading, phytoplankton proliferation, and degraded light climate. The sizeable diurnal variability in AGB (Fig. 9d) appears to be an artifact of production/respiration formulations that are based on seasonal responses to environmental forcing, rather than diurnal responses to solar irradiance. Future modifications could attenuate this variability by utilizing daily averaged environmental forcing, or modifying the frequency of biomass updating.

The modelling framework developed in this work can be used to create hypothetical scenarios to estimate future environmental responses. For example, we ran the model setup of West Falmouth Harbor described in section 3.2 with no nitrate loading, to simulate a hypothetical scenario where the groundwater input has no influence from the wastewater treatment plant (unimpacted past or future scenario). The elimination of nitrate loading results in negligible changes in the outer harbor, but greatly reduces chlorophyll and light attenuation in the inner harbor (Fig. 10a,b), while increasing the near-bottom PAR (Fig. 10c). Peak AGB responds to the decreased chlorophyll and increased light attenuation with an increase in the inner harbor (Fig. 10d). This implementation represents an incremental improvement to the prior modelling exercise (Ganju et al., 2012 and del Barrio et al., 2014), because the interaction between SAV and the nitrogen pools are explicitly accounted for. For example, this model can now be used to test how changes in seagrass coverage influence nitrogen retention within the estuary, or export to the coastal ocean. Further, the introduction of seagrass kinetics will allow for investigation of water column oxygen budgets with and without seagrass, under present and future scenarios.

### 4.3. Model evaluation in West Falmouth Harbor

In order to qualitatively evaluate the seagrass growth model, we have compared the modeled results with observations by del Barrio et al. (2014) that measured the extent of seagrass coverage in West Falmouth Harbor (red outline in Fig. 11). The field data is only available for the northern region of WFH where the model-data comparisons are performed. The model results are compared by extracting the peak above ground biomass (AGB) on 14th day of the simulation and normalized with the initial above ground biomass. The ratio of AGB/AGB$_{initial}$ is considered as a representative of seagrass growth. We assume that for AGB/AGB$_{initial}$ > 1, there is a potential for seagrass growth and for AGB/AGB$_{initial}$ <1, the conditions are unfavorable for seagrass growth. In fig 11, the model and field data show a 89% agreement to determine the seagrass growth or dieback. The western region of outer harbor shows seagrass growth potential and agrees with the extent that the seagrass coverage is observed. In the eastern region, the field data shows no seagrass coverage and the model also predicts potential seagrass dieback. The model predicts seagrass dieback because of nitrate loading from shoreline point sources that leads to increased chlorophyll and light attenuation (figures 8a, b). The model and observations do not compare well in the central basin of outer harbor where the model shows

seagrass dieback potential while the field data shows presence of seagrass. In the central basin, the field data shows the presence of seagrass while its density remains low in this region. On the other hand, the modelled seagrass suffers dieback due to the bathymetric controls in the deeper central basin (decreased near-bottom PAR Fig. 8c).

Direct estimates of above ground SAV biomass have also been recently made in West Falmouth Harbor (Hayn et al., unpublished data). Although these measurements were not made during the same year as our simulations (measurements in 2006, 2007, 2013; model 2010), the mean above ground biomass measured in the outer harbor of 49.5 (June 21-July 6 2006), 45.3 (June 6-19 2007), and 41.5 g C m$^{-2}$ (July 15-19 2013) is consistent with the range of model simulations during a comparable period (July 2-19) in the outer (28.1 to 51.1 g C m$^{-2}$) and middle (14.9 to 37.4 g C m$^{-2}$) harbors. The July 2-19 model range of 45.7 to 156.3 mmol N m$^{-2}$ across the middle and outer harbor is also consistent with annual mean *Z. marina* biomass (10-88 mmol N m$^{-2}$) reported in nearby shallow systems on Cape Cod (Hauxwell et al. 2003) assuming a literature-based average that above ground SAV biomass is 1.5% N. The range in the model is computed based on the minimum and maximum values of AGB during the 18 day simulation period.

### 4.4 Limitations of SAV growth model and Future Work

While this modelling approach represents an advance in modelling coupled biophysical processes in estuaries, there are limitations that must be addressed in future work:

1. The modelling of SAV dieback/growth scenarios may require long-term simulations on decadal timescales (Carr et al., 2018). However, the short model time step limits the duration of such simulations. The time step size is of the order of seconds (typical of 3-D ocean models) and this combined with the fact that the presence of SAV in the hydrodynamic model further limits time step size (due to hydrodynamic stability constraints); overall limits the applicability of the model to be utilized from monthly to annual time scales at this juncture.

2. The biomass equations described in Section 2.3 are formulated for seasonal time scales and are being used in the model implementation at every ocean model time step. This leads to large daily variations in above and below ground biomass that do not likely occur in the environment, although diel variations on SAV growth have been measured in situ (Kemp et al. 1987). Hence, with the current formulations, the output from the biomass model needs to be analyzed as a daily averaged quantity.

3. The current implementation of the SAV growth model is limited to only one SAV species. However, it should be extended to include multiple SAV species to investigate competition under variable salinity and to make the model applicable to a wider variety of locations.

### 5 Conclusions

The present study adds to the open source COAWST modelling framework by implementing a SAV growth model. Based on the change in SAV biomass (above ground, below ground) and epiphyte biomass, SAV density and height evolve in time and space and directly couple to three-dimensional water-column biogeochemical, hydrodynamic, and sediment transport models. SAV biomass is computed from temperature, nutrient loading and light predictions obtained from coupled hydrodynamics (temperature), bio-geochemistry (nutrients) and bio-optical (light) models. In exchange, the growth of SAV sequesters or contributes nutrients from the water column and

sediment layers. The presence of SAV modulates current and wave attenuation and consequently affects modelled sediment transport and fate. The resulting modelling framework provides a two-way coupled SAV-biogeochemistry-hydrodynamic and morphodynamic model. This allows for the simulation of the dynamic growth and mortality of SAV in coastal environments in response to changes in light and nutrient availability, including SAV impacts on

5     sediment transport and nutrient, carbon, and oxygen cycling. The implementation of the model is successfully tested in an idealized domain where the introduction of sediment in the water column (SSC) at one end of the domain provides sub-optimal light conditions that causes SAV dieback in that region. The model was applied to the temperate estuary of West Falmouth Harbor, where simulations show the coupled effect of enhanced nitrate loading in the inner harbour leading to poor light conditions for the SAV to grow; thus modelling the physical effect of eutrophication

10    leading to the loss of SAV habitat. Among other applications, in future, the model will be used assess the effects of sea level rise scenarios that limit light availability and potentially cause the loss of SAV habitat.

**6 Code availability**

The implementation of the SAV growth model has been implemented in the Coupled Ocean Atmosphere Waves Sediment-Transport Modeling System (COAWST v3.4). This particular version is available for download at: https://www.sciencebase.gov/catalog/item/5f15d69082cef313ed81996a. Users are encouraged to download COAWST

distributed through the USGS code archival repository. It is available for download on https://code.usgs.gov/coawstmodel/COAWST. The COAWST distribution files contain source code derived from ROMS, SWAN, WRF, MCT and SCRIP, along with the Matlab code, examples and a User's Manual.

The major code development that was done for this project is contained within the COAWST folder on the following path. "https://code.usgs.gov/coawstmodel/COAWST/blob/master/ROMS/Nonlinear/Biology/"

This folder contains several methods of computing water column biogeochemistry. Other than the I/O component of our implementation, the algorithmic development in this study only modifies two files on this path: "estuarybgc.h" and "sav_biomass.h". The file "sav_biomass.h" contains all the newly added equations for the growth of SAV based on the nutrient loading in the water column. The forcings to the SAV growth model (temperature, light, nutrient availability, exchanges nutrients, detritus, dissolved inorganic carbon, and dissolved oxygen) are provided through the file "estuarybgc.h" that calls

"sav_biomass.h". The file "estuarybgc.h" solves for the water column biogeochemistry and was based on existing modelling framework developed by Fennel et al. (2006) (also coded as "fennel.h").

Other important paths that existed in the framework prior to the current modeling effort but are being used in the modeling process include:

1. "https://code.usgs.gov/coawstmodel/COAWST/blob/master/ROMS/Nonlinear"-

The main kernel of the 3-D non-linear Navier-Stokes equations is contained within this part and links all the submodels: biological, vegetation and sediment models.

2. "https://code.usgs.gov/coawstmodel/COAWST/blob/master/ROMS/Nonlinear/Vegetation/"

The kernals that account for seagrass-hydrodynamics interactions.

3. "https://code.usgs.gov/coawstmodel/COAWST/blob/master/ROMS/Nonlinear/Sediment/"

The kernals that account for sediment transport.

**7 Data availability**

The model data was released as per the USGS model data release policy and separate digitial object identifiers were created as part of the release (https://www.usgs.gov/products/data-and-tools/data-management/data-release). For each of the model

data releases, separate landing pages are constructed and the model data can be either accessed through thredds server or directly downloaded in netcdf format. The model output from the idealized test case simulation (Kalra and Ganju, 2019) can be accessed via thredds server or directly downloaded in netcdf format from this link:
https://www.sciencebase.gov/catalog/item/5d3b4d32e4b01d82ce8d77f5

The model output from the West Falmouth Harbor simulation (Ganju and Kalra, 2019) can be accessed via thredds server from

this link: https://www.sciencebase.gov/catalog/item/5d42f064e4b01d82ce8daf41 and the

model output from the West Falmouth Harbor simulation to model the hypothetical future scenario with the elimination of nitrate loading can be accessed via thredds server from this link: : https://www.sciencebase.gov/catalog/item/5d42f08ee4b01d82ce8daf49

Both the West Falmouth Harbor simulations can be directly downloaded in netcdf format from this link: https://www.sciencebase.gov/catalog/item/5d8b964be4b0c4f70d0bbad8

## 8 Author contribution

T. S. Kalra implemented the SAV growth model in the COAWST framework. J. Testa provided guidance on the mechanistic

processes affecting the growth of SAV from biomass parameterizations and developed linkages between the SAV growth model and the water-column biogeochemical model. N. K. Ganju developed the test case and the realistic domain case. T. S. Kalra and N. K. Ganju performed the data analysis from the output of the test cases and were responsible for model data release. The manuscript was prepared with contributions from all co-authors.

## 9 Disclaimer

This draft manuscript is distributed solely for purposes of scientific peer review. Its content is deliberative and pre-decisional, so it must not be disclosed or released by reviewers. Because the manuscript has not yet been approved for publication by the U.S. Geological Survey (USGS), it does not represent any official USGS finding or policy. Any use of trade, firm, or product names is for descriptive purposes only and does not imply endorsement by the U.S. Government.

## 10 Acknowledgements

We thank Dr. Joel Carr at the US Geological Survey Patuxent Wildlife Research Center in Beltsville, Maryland for providing his feedback to improve the quality of the paper. We thank the reviewers for their careful reading of our manuscript and their insightful comments and suggestions. We would also like to thank Melanie Hayn, Robert Howarth, Roxanne Marino, and

Karen McGlathery for sharing *Z. marina* biomass measurements from West Falmouth Harbor collected with funding from the National Science Foundation and Woods Hole SeaGrant to Cornell University. All the contour plots are generated using the "cmocean" package developed by Thyng et al., 2016. The authors would thank VeeAnn Cross and associated staff at USGS that helped in model data release in a timely manner. This is University of Maryland Center for Environmental Contribution # XXXX.

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

| Model variables | Equivalent symbol in source code | Description of terms |
|---|---|---|
| $\alpha$ | $AGB$ | Above ground biomass |
| $\beta$ | $BGB$ | Below ground biomass |
| $\gamma$ | $EPB$ | Epiphyte biomass |
| $\mu_s$ | $ua_{SAV}$ | Maximum potential growth rate (SAV) |
| $\rho_s$ | $pp_{SAV}$ | Primary production (SAV) |
| $\varphi_s$ | $llmt_{SAV}$ | Light availability (SAV) |
| $\vartheta_s$ | $nlmt_{SAV}$ | Nutrient limitation (SAV) |
| $\lambda_s$ | $lambda_{SAV}$ | Self-shading term (SAV) |
| $\delta_s$ | $agar_{SAV}$ | Active respiration (SAV) |
| $\varepsilon_s$ | $agbr_{SAV}$ | Above ground basal respiration (SAV) |
| $\omega_{s,1}$ | $agm_{SAV}$ | Above ground biomass mortality (SAV) |
| $\omega_{s,2}$ | $bgm_{SAV}$ | Below ground biomass mortality (SAV) |
| $\theta$ | $PAR$ | Photosynthetic Active Radiation (PAR) |
| $\sigma_{s,1}$ | $agbg_{SAV}$ | Translocation of above ground biomass to below ground biomass (SAV) |
| $\sigma_{s,2}$ | $bgbg_{SAV}$ | Translocation of below ground biomass to above ground biomass (SAV) |
| $\mu_e$ | $ua_{EPB}$ | Maximum potential growth rate (EPB) |
| $\rho_e$ | $pp_{EPB}$ | Primary production (EPB) |
| $\varphi_e$ | $llmt_{EPB}$ | Light availability (EPB) |
| $\vartheta_e$ | $nlmt_{EPB}$ | Nutrient limitation (EPB) |
| $\lambda_e$ | $lambda_{EPB}$ | Self-shading term (EPB) |
| $\varepsilon_{e,1}$ | $aresp_{EPB}$ | Active respiration (EPB) |
| $\varepsilon_{e,2}$ | $bresp_{EPB}$ | Basal respiration (EPB) |
| $\omega_e$ | $mort_{EPB}$ | Mortality (EPB) |
| $\tau_{e,1}$ | $grz_{EPB}$ | Loss of EPB due to grazing |
| $\tau_{e,2}$ | $epb_{slgh}$ | Epiphyte biomass due to the SAV sloughing loss |
| $t$ | $dtdays$ | Time step in days |

**Table 1: SAV growth model variable descriptions**

| Input parameter | Equivalent symbol in model | Description | Default value | Units |
|---|---|---|---|---|
| $g_s$ | $scl$ | SAV growth fraction | 0.03 | None |
| $r_s$ | $arc$ | SAV active respiration coefficient | 0.01 | dtdays$^{-1}$ |
| $T_{s,o}$ | $T_{OPT}$ | Optimum SAV growth temperature | 15.0 | °C |
| $\lambda_{s,max}$ | $\lambda_{SAV,max}$ | Self-shading parameter for SAV leaves (maximum AGB) | 475.0 | millimoles N m$^{-2}$ |
| $l_s$ | $klmt$ | Half-saturation for light limitation for SAV | 100.0 | E m$^{-2}$ s$^{-1}$ |
| $n_{s,1}$ | $kn_t$ | Half-saturation for nutrient limitation for plant roots | 100.0 | millimoles |
| $n_{s,2}$ | $kn_{wc}$ | Half-saturation for nutrient limitation for plant leaves | 5.71 | millimoles |
| $p_s$ | $arsc$ | Maximum fraction of photosynthesis, SAV respiration | 0.1 | None |
| $c_s$ | $bsrc$ | Maximum fraction of SAV BGB biomass respired | 0.0015 | None |
| $h_s$ | $rc$ | SAV basal respiration coefficient (AGB and BGB) | 0.069 | dtdays$^{-1}$ |
| $m_\alpha$ | $km_{ag}$ | SAV AGB mortality rate (sloughing) | 0.0005 | dtdays$^{-1}$ |
| $m_\beta$ | $km_{bg}$ | SAV BGB mortality rate | 0.005 | dtdays$^{-1}$ |
| $g_e$ | $scl_{EPB}$ | Epiphyte growth fraction | 0.2 | None |
| $r_e$ | $arc_{EPB}$ | Epiphyte active respiration coefficient | 0.0633 | dtdays$^{-1}$ |
| $T_{e,o}$ | $T_{EPB,opt}$ | Optimum growth temperature for epiphytes | 25.0 | °C |
| $\lambda_{e,max}$ | $\lambda_{EPB,max}$ | Self-shading parameter for epiphytes (maximum EPB) | 475.0 | millimoles N m$^{-2}$ |
| $l_e$ | $kl_{EPB}$ | Half-saturation for light limitation for SAV | 50.0 | E m$^{-2}$ s$^{-1}$ |
| $n_e$ | $kn_{EPB}$ | Half-saturation for nutrient limitation for SAV | 10.0 | millimoles |
| $p_e$ | $arsc_{EPB}$ | Maximum fraction of photosynthesis, EPB active respiration | 0.01 | None |
| $c_e$ | $brsc_{EPB}$ | Maximum fraction of SAV BGB biomass respired | 0.0015 | None |
| $h_e$ | $rc_{EPB}$ | SAV basal respiration coefficient (AGB and BGB) | 0.069 | dtdays$^{-1}$ |
| $m_e$ | $kmort_{EPB}$ | Mortality rate for epiphytes if no sloughing | 0.001 | dtdays$^{-1}$ |
| $z_{e,max}$ | $grz_{EPB,max}$ | Maximum grazing rate on epiphytes | 0.1 | dtdays$^{-1}$ |
| $z_e$ | $grzk_{EPB}$ | Grazing coefficient on epiphytes | 0.01 | None |
| $k_1$ | $kd_{trans}$ | Downward translocation coefficient | 0.1 | None |
| $k_2$ | $ku_{trans}$ | Upward translocation coefficient | 0.02 | None |
| $m_f$ | $mx_{frc}$ | Upward translocation coefficient | 0.02 | None |

**Table 2: SAV growth model input parameter descriptions and their values**

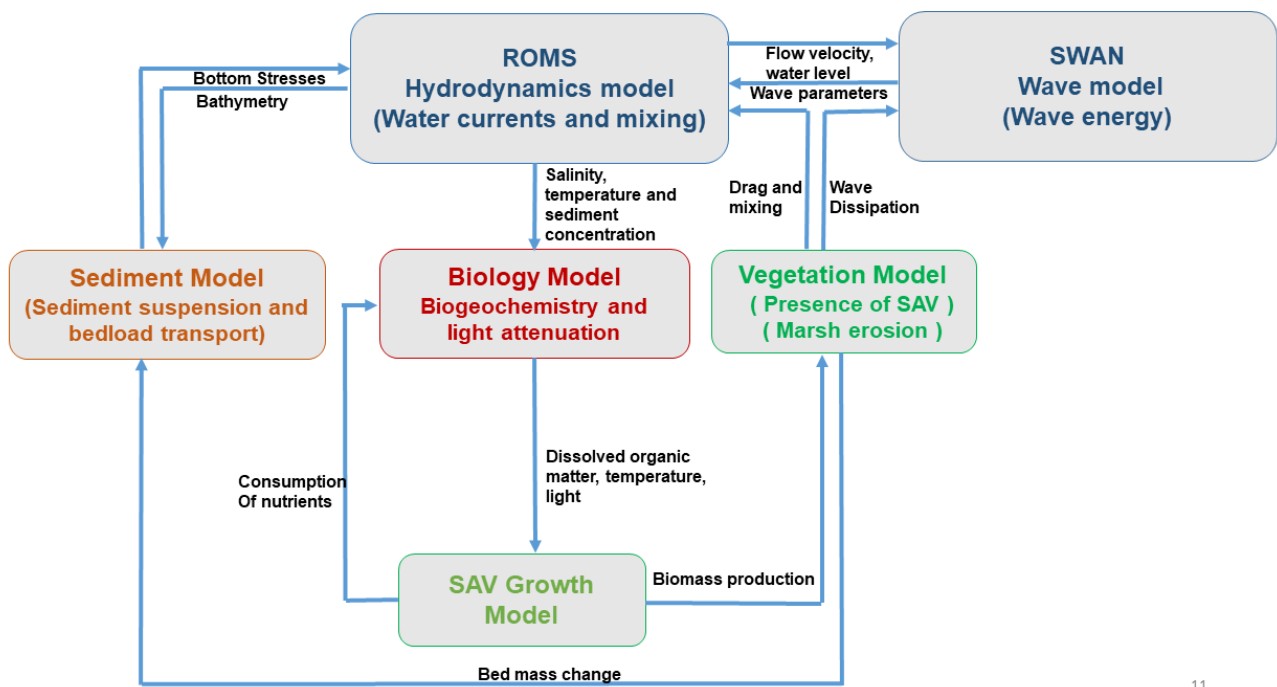

**Figure 1: Schematic showing the coupling of SAV growth module implementation in COAWST model.**

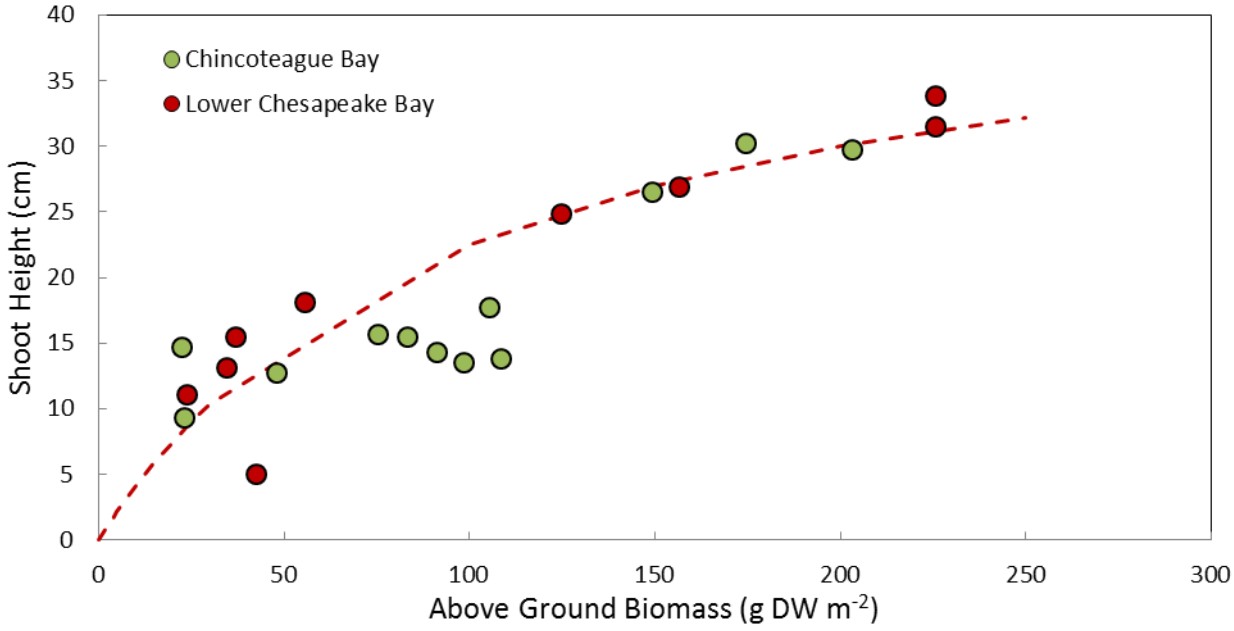

**Figure 2: Empirical relationships between above ground biomass and SAV shoot height for Z. marina populations in polyhaline regions of Chesapeake Bay and Chincoteague Bay. Data from Moore et al. 2004 and Ganju et al. 2018.**

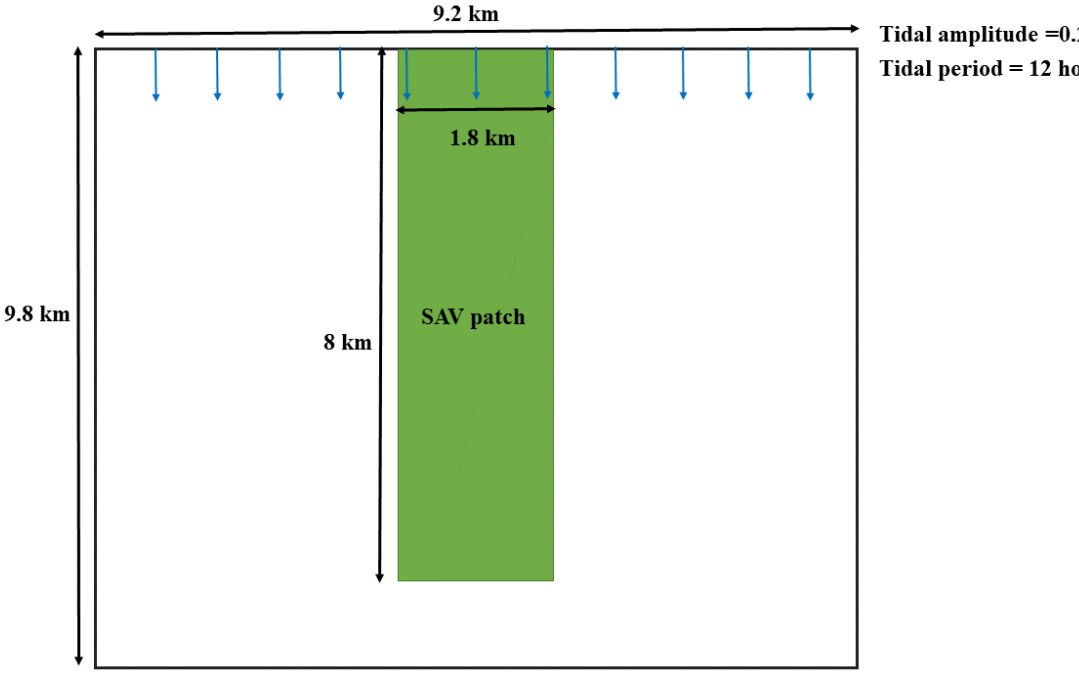

**Figure 3: Planform view of the idealized test domain simulation.**

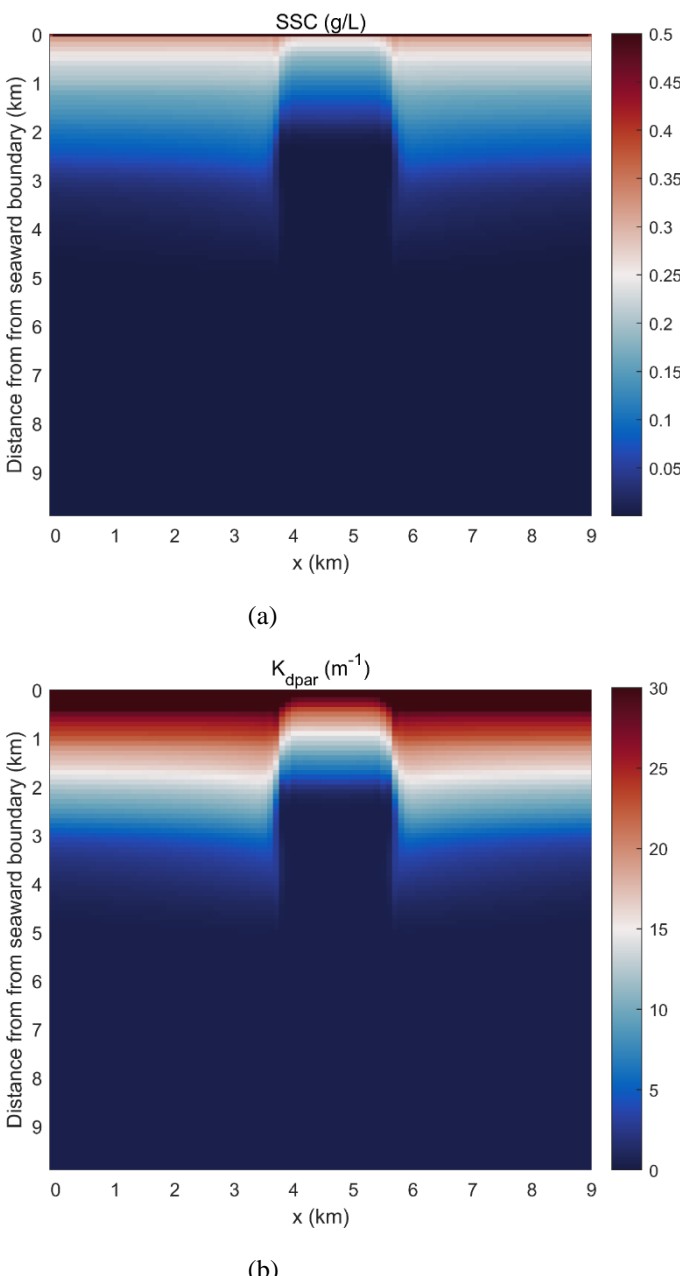

(a)

(b)

Figure 4: Planform view of (a) depth-integrated SSC, (b) light attenuation averaged over the last day of the simulation in the idealized domain.

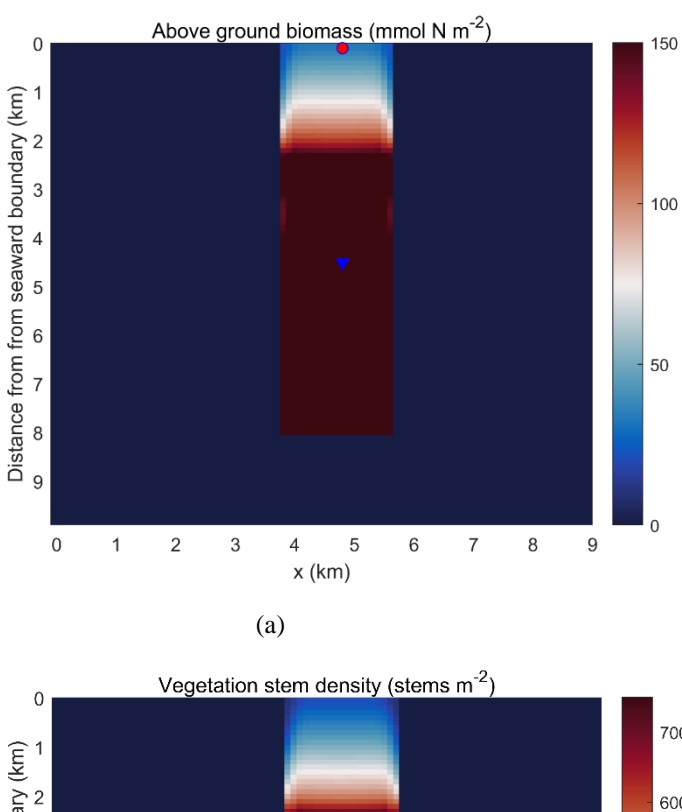

(a)

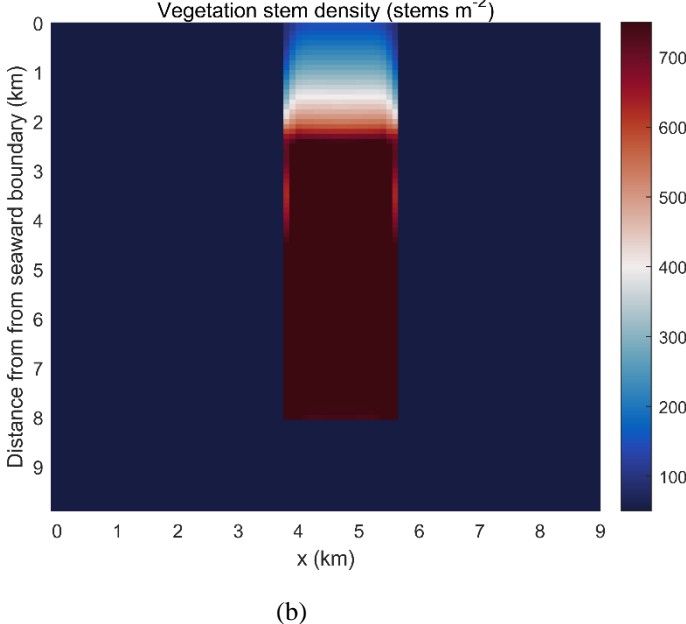

(b)

Figure 5: Planform view of (a) above ground biomass and (b) vegetation stem density averaged over the last day of the simulation in the idealized domain. Red dot and blue triangle represent two points that are located at 0.1 km and 4.5 km into the SAV bed respectively.

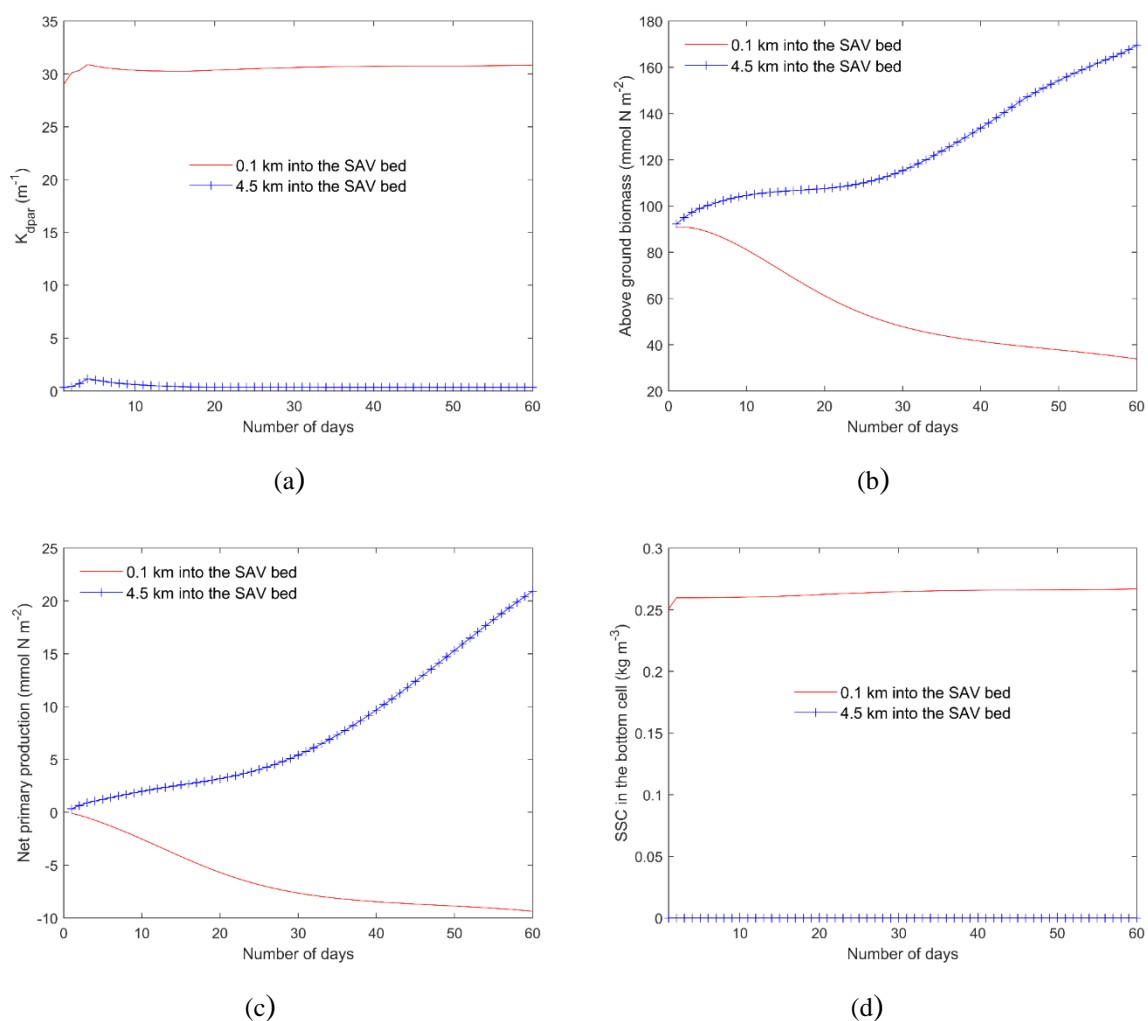

5                              (c)                                   (d)

**Figure 6: Time-series of a) light attenuation, b) above ground biomass, c) net primary production of SAV $(\rho_s - \delta_s - \varepsilon_s)$, and d) SSC in the bottom cell averaged every day from the two locations identified in Fig. 5a.**

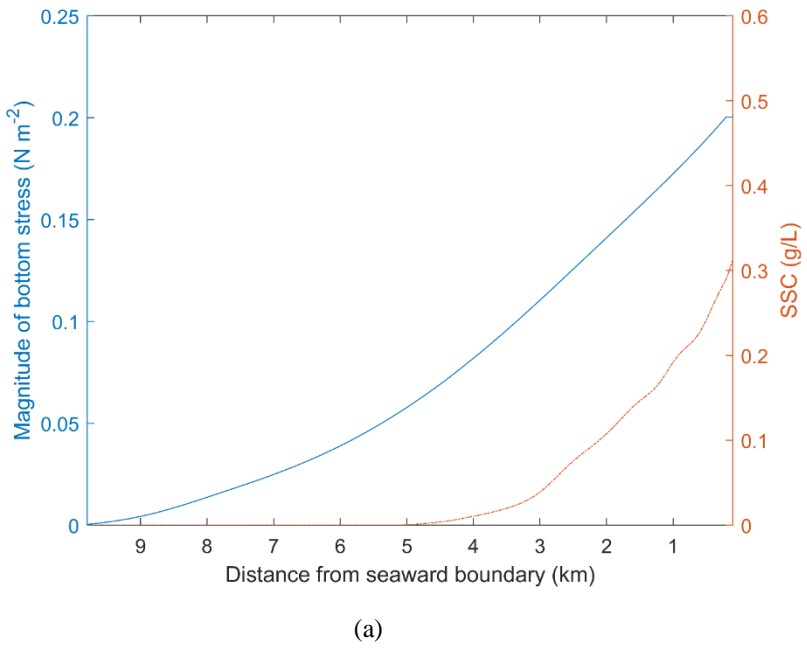

(a)

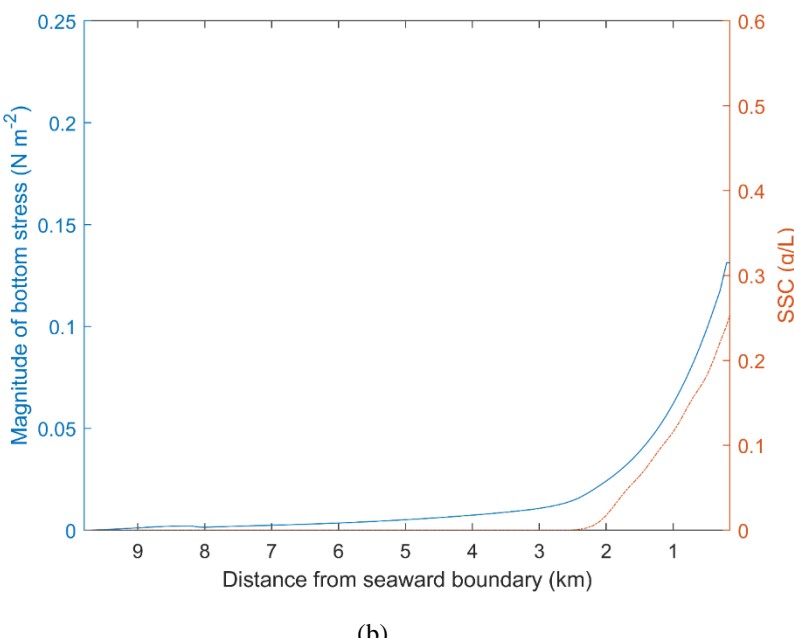

5                                    (b)

**Figure 7. Magnitude of bottom stress (left) and depth-integrated SSC (right) at the end of the simulation plotted along the y axis of the idealized domain at two locations, including one outside (x=1.8 km; panel a) and one inside the SAV bed (x=4.8 km, panel b)**.

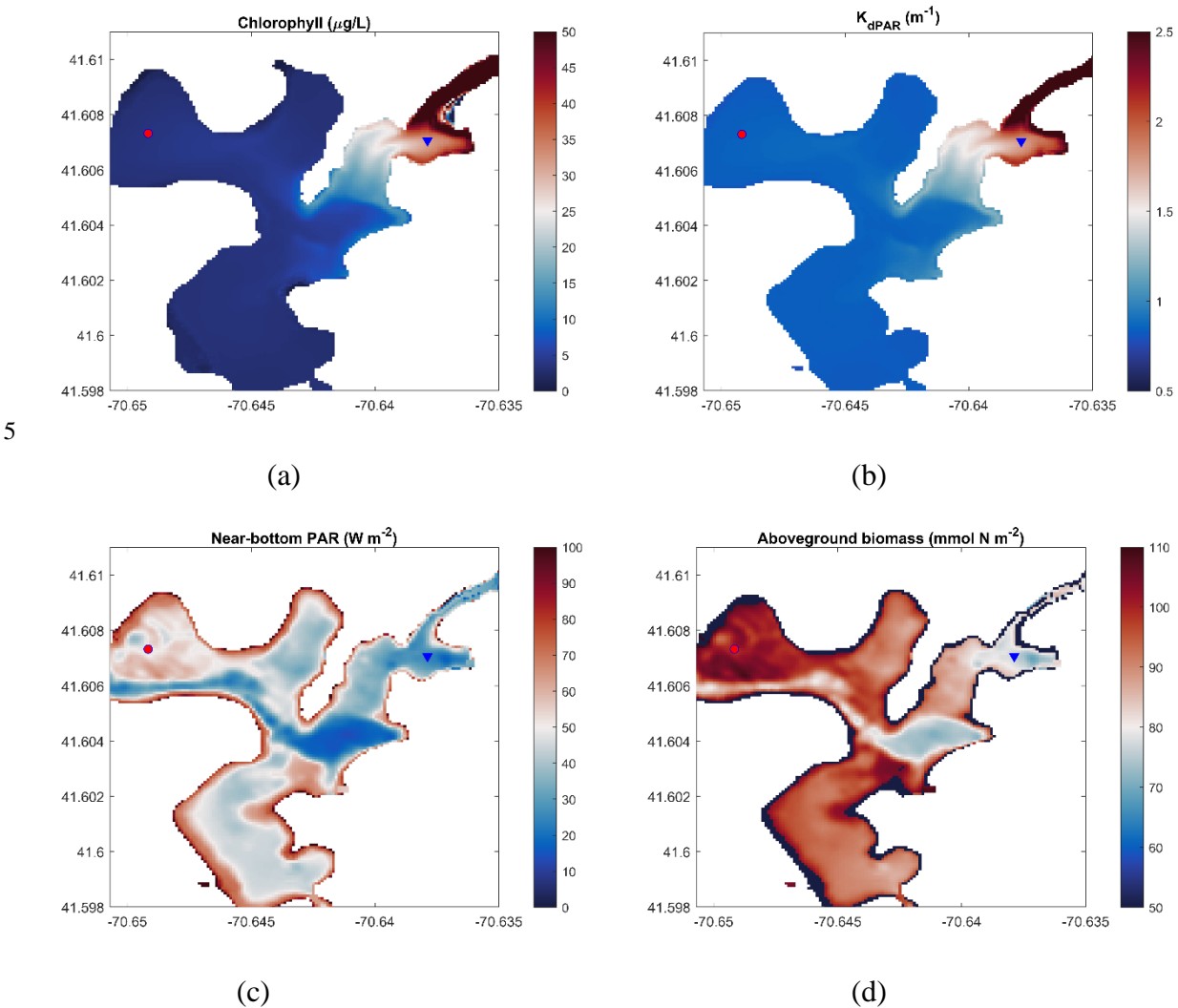

(a)

(b)

(c)

(d)

**Figure 8. Mean over 22 days of a) depth-averaged chlorophyll, b) light attenuation, c) near-bottom PAR, and d) peak above ground biomass at day 14 of the simulation. Red circle indicated outer harbor (left) and blue triangle indicated inner harbor (right) points for time-series data in Figure 9.**

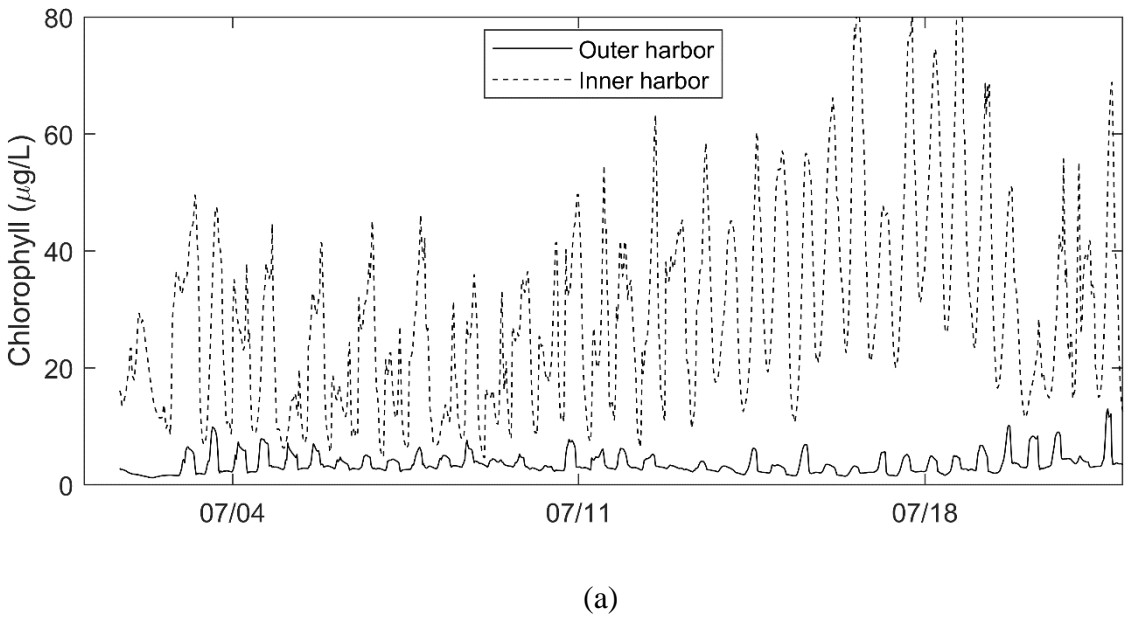

(a)

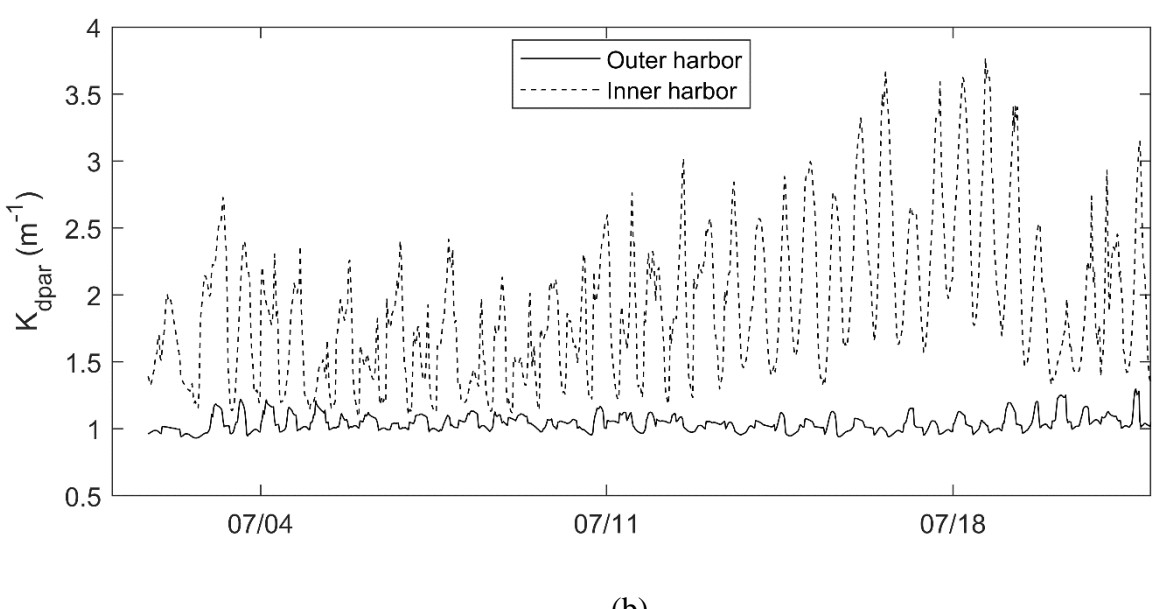

(b)

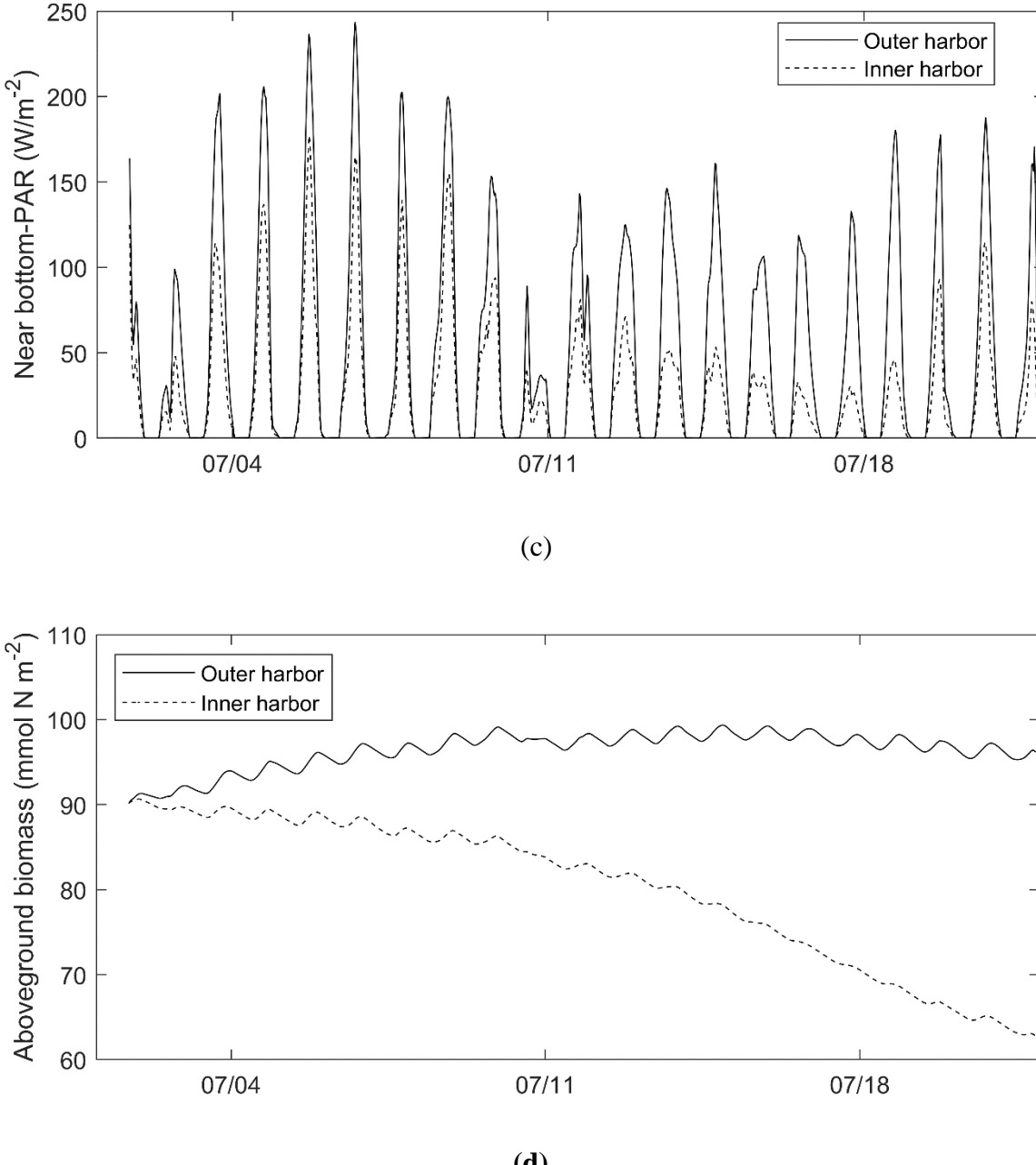

(c)

(d)

**Figure 9. Time-series of a) chlorophyll, b) light attenuation, c) near-bottom PAR, and d) above ground biomass from outer and inner harbor locations identified in Figure 6.**

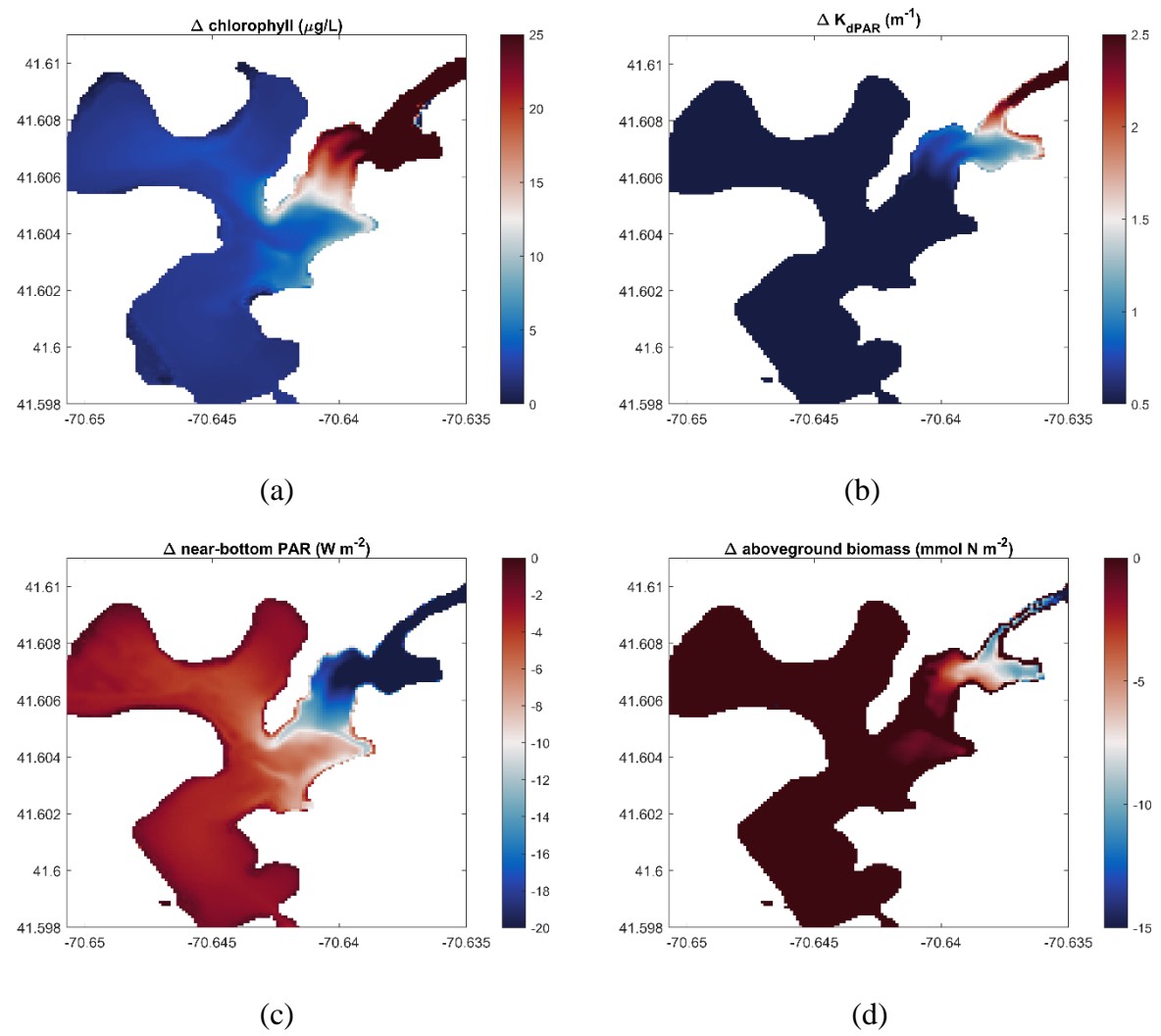

**Figure 10. Change in outcomes between impacted and non-impacted scenario (nitrate loading scenario – no loading scenario). Difference in mean over 22 days of (a) depth-averaged chlorophyll, (b) light attenuation, (c) near-bottom PAR, and (d) peak above ground biomass at day 14 of the simulation**.

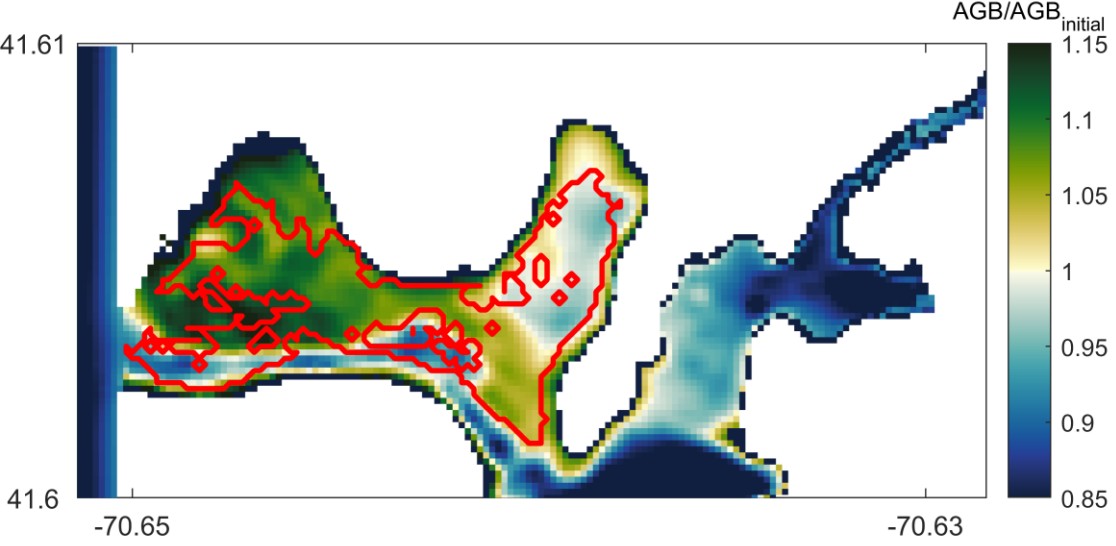

**Fig 11: Modeled AGB/AGB**$_{initial}$ **(above ground biomass) distribution compared with field data showing seagrass coverage extent (red solid line). Values of AGB/AGB**$_{initial}$ **> 1 represent seagrass growth potential and below 1 indicate potential seagrass decline at day 14 of the simulation.**

