# Peer review of "Development of a Submerged Aquatic Vegetation Growth Model in a Coupled Wave-Current-Sediment Model (COAWST v3.4)"

_Geoscientific Model Development, 2018_

## Referee Comment (RC1) · Jon Hill (Referee) · 1 Jul 2019

Summary:

This paper details a new seagrass model incorporated into COAWST that includes two-way interactions with both physical and biological processes included in the model. The paper describes the complex set of equations used in the seagrass model and shows the model performance on two examples: an idealised case and a more realsitic case. In both examples, the effects of two-way coupling is shown , but there is a focus on the

biological reactions, rather than the impact of seagrass changes on hydrodynamics.

Overall the paper is generally well-written and clear, but lacks some sort of validation or verification of the sea grass model. My main criticism of the paper is that this verification is lacking and it is therefore difficult to ascertain if the model works compared to some lab or case study. Whilst the two examples seem sensible it does not show proper functioning of the code. I didn't attempt to run the code in question as part of the review, but I couldn't actually find the seagrass model in the code repository easily, so could not even check equation as written in the paper match the code.

Requested changes:

======================

Major:

- Add some sort of verification. I assume this has been done as part of some sort of testing infrastructure, so should be trivial to add to the paper.

- Check code availability and make it clearer which parts of COAWST are part of this paper. As the editor has indicated, a Zenodo archive, coupled with some indication of which code this paper refers to would be a great help.

- Equations in 2.2 are very difficult to read with "words" being used as symbols in a lot of cases; especially when "lim" is used in a symbol it makes it difficult to know of this is the mathematical limit of or a symbol at a glance. Symbols such as lambda_SAV-max (eq 3) should be altered to remove operation symbols from them. There are also symbols such as kl. Is this k * l or a symbol kl? I would recommend the use of single symbols where possible and remove as many "words" as possible. Same applies to table 1.

Minor:

- The abstract has a few complex sentences, e.g. "Recent observational studies..."

**[GMDD](GMDD)**
[Figure]

(lines 11-13) and "Modelled SAV biomass is represented..." (lines 16-17), etc. Best to rewrite into simpler sentences or make them clear - the use of lists, with multiple "and"s make it unclear to work out what is being referred to at times.

- Line 25, pg 2 - extra () round reference

- Line 26, pg 11 - typo: "diel"

- Figure 3 - remove orientation axes. It's plan view, so z isn't on!

- Fig 4 - Capital letters in axes title

- Fig 5 - triangle and dot not explained in caption. Capital letters in axes titles

- Fig 6 - Capital letters in axes titles. Remove "Figure" from sub captions

- Fig 7 - Capital letters in axes titles.

- Fig 8 - replace "rainbow/jet" colour scheme with colour-blind friendly scheme. See here for examples: https://matplotlib.org/cmocean/

- Fig 10 - as above.
* * *

---

## Referee Comment (RC2) · David Ham (Referee) · 24 Feb 2020

This review is being conducted by me as topical editor for this manuscript. This is an unusual and somewhat unfortunate occurrence which has been caused by two reviewers in series failing to produce their reports.

This manuscript introduces a new vegetation model in a coastal ocean model. It is within scope for GMD and is potentially a valuable contribution, however at this stage the manuscript is let down by rather serious deficiencies in the description of the model,

and in its verification and evaluation. These will need to be corrected before a revised manuscript can be accepted.

**1 Mathematical notation**

It is unconventional for a review to start with something this technical, however in this case the highly unconventional mathematical notation makes the equations so difficult to read that the meaning is severely impaired.

1. Mathematical symbol names should be single letters (Latin, Greek, or potentially from another alphabet if really needed). Using multi-letter names creates confusion about what is a variable name and what is a multiplication of symbols. This is a convention that very much also holds in the marine biogeochemistry modelling community, for example the NPZD model is named after the conventional (single letter) symbol names for its four prognostic quantities).

2. If it is necessary or useful to use a multi-letter subscript or superscript to further identify a variable, then this should be typeset in upright letters to avoid the confusion with a product of symbols. Using LaTeX, this can be achieved with `\mathrm`, for example $T_{\mathrm{opt}}$ is written as `$T_{\mathrm{opt}}$`.

3. $\exp$ is the exponential function, it takes its argument in round brackets and not as an index. $e$ is a number, the base of natural logarithms, and can be exponentiated by writing an index. The current mix of these two notations, for example in equation 2, is at best confusing and at worst meaningless.

4. Mathematical function names are typeset upright and usually use lower case letters, for example $\exp$, $\min$ (`$\exp$`, and `$\min$` respectively).

**[GMDD](GMDD)**

[Figure]

5. Double subscripts should be avoided where possible. If they are unavoidable then they should not be separated by a hyphen, because a horizontal line universally means subtraction. A comma, possibly augmented by brackets of some type, would be a better choice.

**2  Equations and discretisation**

The introduction to section 2.2 claims that the remainder of the section will introduce the equations solved. In fact, we are only treated to a disconnected set of source terms for an unspecified set of equations.

Please provide the full set of differential equations being solved, before going into detail about the definition of the terms. In addition, the equations are clearly being solved numerically, so a complete model description also requires the inclusion of the discretisation used, and how the resulting discrete linear or nonlinear system is solved.

**3  Verification and evaluation**

There is effectively no verification or validation of the model. The test cases provided are purely descriptive: the model is run and the authors describe what happened. This does not provide suitable evidence either that the model is correctly implemented, or that it is realistic. The usual way of demonstrating the former would be using the method of manufactured solutions (MMS) to create artificial analytical solutions to the system, and then demonstrating convergence to them at the expected rate. For more information on MMS see Farrell et al. (2011) section 4.1 (https://doi.org/10.5194/gmd-4-435-2011).

In order to provide some level of evaluation of the model, it would be necessary to

present a qualitative or quantitative comparison of the model to an external reference. The external reference might be directly with observational data, or might be with the results of another well-evaluated model. In any event, an external comparator is absolutely necessary.

**[GMDD](

---

## Author Response (AR1)

Dear Editor,

On behalf of my co-authors, I am pleased to resubmit the manuscript "Development of a Submerged Aquatic Vegetation Growth Model in a Coupled Wave-Current-Sediment-Transport Modeling System (COAWST v3.4)" to be considered for publication in Geophysical Model Development.

The manuscript covers the description of a novel method that models the growth of submerged aquatic vegetation (SAV) allowing it to dynamically grow or dieback based on the temperature, light and nutrient availability in the water column. The dynamic simulation of the SAV allows the plants to both respond to and cause change in water column and sediment bed properties, hydrodynamics and sediment transport (two-way feedback). We think that the manuscript has the potential to be useful for the coastal modeling community in general. More specifically, it can help scientists trying to provide informed advice to coastal managers about the complex feedbacks between nutrient loading in water column, water quality issues and SAV growth in the development of a green infrastructure. The model can also be used to assess the effects of sea level rise scenarios that limit light availability and potentially cause the loss of SAV habitat.

Following the publishing process at USGS, the discussion paper was internally reviewed by a USGS scientist. After the suggestions of the internal reviewer and external reviewers were incorporated, it was reviewed to ensure that the standards of USGS policy of peer review were met.

The issues raised with the previous submission have been addressed following the Reviewers' guidance. The reviewers major concern was to address the issue of verification/evaluation of the developed model and describing the integration of the code in the larger framework. Along with adding the equations that describe the integration of the newly developed model in the existing software, we have added additional text and figures to present model verification with available field studies. We have incorporated all the other changes based on the reviewers' suggestions and addressed their comments in details. Please find the documents that address the suggestions and comments from all the referees. The sub-sections in the document contain the following:

Sincerely,
Tarandeep

**1.1 Comments from Referee 1**

This paper details a new seagrass model incorporated into COAWST that includes two way interactions with both physical and biological processes included in the model. The paper describes the complex set of equations used in the seagrass model and shows the model performance on two examples: an idealised case and a more realsitic case. In both examples, the effects of two-way coupling is shown , but there is a focus on the biological reactions, rather than the impact of seagrass changes on hydrodynamics. Overall the paper is generally well-written and clear, but lacks some sort of validation or verification of the sea grass model. My main criticism of the paper is that this verification is lacking and it is therefore difficult to ascertain if the model works compared to some lab or case study. Whilst the two examples seem sensible it does not show proper functioning of the code. I didn't attempt to run the code in question as part of the review, but I couldn't actually find the seagrass model in the code repository easily, so could not even check equation as written in the paper match the code.

Requested changes:

Major:
- Add some sort of verification. I assume this has been done as part of some sort of testing infrastructure, so should be trivial to add to the paper.

- Check code availability and make it clearer which parts of COAWST are part of this paper. As the editor has indicated, a Zenodo archive, coupled with some indication of which code this paper refers to would be a great help.

- Equations in 2.2 are very difficult to read with "words" being used as symbols in a lot of cases; especially when "lim" is used in a symbol it makes it difficult to know of this is the mathematical limit of or a symbol at a glance. Symbols such as lambda_SAVmax(eq 3) should be altered to remove operation symbols from them. There are also symbols such as kl. Is this k * l or a symbol kl? I would recommend the use of single symbols where possible and remove as many "words" as possible. Same applies to table 1.

Minor:
- The abstract has a few complex sentences, e.g. "Recent observational studies..."(lines 11-13) and "Modelled SAV biomass is represented..." (lines 16-17), etc. Best to rewrite into simpler sentences or make them clear - the use of lists, with multiple "and"make it unclear to work out what is being referred to at times.

- Line 25, pg 2 - extra () round reference

- Line 26, pg 11 - typo: "diel"

- Figure 3 - remove orientation axes. It's plan view, so z isn't on!

- Fig 4 - Capital letters in axes title

- Fig 5 - triangle and dot not explained in caption. Capital letters in axes titles

- Fig 6 - Capital letters in axes titles. Remove "Figure" from sub captions

5   - Fig 7 - Capital letters in axes titles.

- Fig 8 - replace "rainbow/jet" colour scheme with colour-blind friendly scheme. Seehere for examples: https://matplotlib.org/cmocean/

10   - Fig 10 - as above.

**1.2 Comments from Referee 2**

This review is being conducted by me as topical editor for this manuscript. This is an unusual
and somewhat unfortunate occurrence which has been caused by two reviewers in series failing
to produce their reports. This manuscript introduces a new vegetation model in a coastal ocean
model.

It is within scope for GMD and is potentially a valuable contribution, however at this stage the
manuscript is let down by rather serious deficiencies in the description of the model and in its
verification and evaluation. These will need to be corrected before a revised manuscript can be
accepted.

1 Mathematical notation
It is unconventional for a review to start with something this technical, however in this case the
highly unconventional mathematical notation makes the equations so difficult to read that the
meaning is severely impaired.

1. Mathematical symbol names should be single letters (Latin, Greek, or potentially from another
alphabet if really needed). Using multi-letter names creates confusion about what is a variable
name and what is a multiplication of symbols. This is a convention that very much also holds in
the marine biogeochemistry modelling community, for example the NPZD model is named after
the conventional (single letter) symbol names for
its four prognostic quantities).

2. If it is necessary or useful to use a multi-letter subscript or superscript to further identify a
variable, then this should be type set in upright letters to avoid the confusion with a product of
symbols. Using LATEX, this can be achieved with \mathrm, for example Topt is written as
$T_{\mathrm{opt}}$.

3. exp is the exponential function, it takes its argument in round brackets and not as an index. e is
a number, the base of natural logarithms, and can be exponentiated by writing an index. The
current mix of these two notations, for example in equation 2, is at best confusing and at worst
meaningless.

4. Mathematical function names are typeset upright and usually use lower case letters, for
example exp,
 min ($\exp$, and $\min$ respectively).

5. Double subscripts should be avoided where possible. If they are unavoidable then they should
not be separated by a hyphen, because a horizontal line universally means subtraction. A comma,
possibly augmented by brackets of some type, would be a better choice.

2 Equations and discretisation
The introduction to section 2.2 claims that the remainder of the section will introduce the
equations solved. In fact, we are only treated to a disconnected set of source terms for an

unspecified set of equations. Please provide the full set of differential equations being solved, before going into detail about the definition of the terms. In addition, the equations are clearly being solved numerically, so a complete model description also requires the inclusion of the discretisation used, and how the resulting discrete linear or nonlinear system is solved.

3 Verification and evaluation

There is effectively no verification or validation of the model. The test cases provided are purely descriptive: the model is run and the authors describe what happened. This does not provide

10 suitable evidence either that the model is correctly implemented, or that it is realistic. The usual way of demonstrating the former would be using the method of manufactured solutions (MMS) to create artificial analytical solutions to the system, and then demonstrating convergence to them at the expected rate. For more information on MMS see Farrell et al. (2011) section 4.1 (https://doi.org/10.5194/gmd-4-435-2011).

In order to provide some level of evaluation of the model, it would be necessary to present a qualitative or quantitative comparison of the model to an external reference. The external reference might be directly with observational data, or might be with the results of another well-evaluated model. In any event, an external comparator is absolutely necessary.

**2.1 Author response to Referee 1**

The response to the Reviewer's comments are in black while the original comments are in *blue.*
*This paper details a new seagrass model incorporated into COAWST that includes two way*
*interactions with both physical and biological processes included in the model. The paper*
*describes the complex set of equations used in the seagrass model and shows the model*
*performance on two examples: an idealised case and a more realsitic case. In both examples, the*
*effects of two-way coupling is shown, but there is a focus on the biological reactions, rather than*
*the impact of seagrass changes on hydrodynamics. Overall the paper is generally well-written*
*and clear, but lacks some sort of validation or verification of the sea grass model. My main*
*criticism of the paper is that this verification is lacking and it is therefore difficult to ascertain if*
*the model works compared to some lab or case study. Whilst the two examples seem sensible it*
*does not show proper functioning of the code. I didn't attempt to run the code in question as part*
*of the review, but I couldn't actually find the seagrass model in the code repository easily, so*
*could not even check equation as written in the paper match the code.*

The reviewers correctly point that the focus of the paper is on the biological growth of SAV and
how the two way coupling is shown to work in an idealized and realistic model domain.

The impact of seagrass changes on hydrodynamics (seagrass-hydrodynamics coupling) in the
model were detailed in an earlier work by Beudin et al. (2017) and also applied later in
Chincoteague Bay (Beudin et al. 2017) . In this work, we have focused on the implementation of
the seagrass growth model that also allows for the operation of a two-way coupled framework
between different modeling components (seagrass, hydrodynamics, biology and sediment
dynamics). We have added the following conclusions to clarify that the impact of seagrass on
hydrodynamics were studied in a previous study (Page 2 in Introduction from Line 23 onwards).

"Recently, Beudin et al. 2017 implemented the physical effects of SAV in a vertically
varying water column through momentum extraction, vertical mixing as well as accounting for
wave dissipation due to vegetation. These processes were implemented within the open source
COAWST (Coupled-Ocean-Atmospheric-Wave-Sediment Transport) modelling system that

couples the hydrodynamic model (ROMS), the wave model (SWAN) and the Community Sediment Transport Modelling System (CSTMS) (Warner et al., 2010). Through this effort, the COAWST framework accounted for the coupled seagrass-hydrodynamics interactions. The model reproduced the turbulent shear stress across the canopy interface and peaked at the top of

5   the canopy similar to the observations of Ghisalberti and Nepf (2004, 2006). The presence of seagrass patch led to a reduced shear-scale turbulence within the canopy and an enhanced wake-scale generated turbulence. For more details on the impact of seagrass on hydrodynamics, readers are referred to Beudin et al. 2017."

10   The main focus of this paper is to implement a seagrass growth model and couple various existing components seagrass, hydrodynamics, biological and sediment dynamics. We have added verification of the seagrass growth model with available observations in a new section (Section 4.3).

The following are a response to the reviewers major comments.

15   *Major comments:*

*Comment 1: Add some sort of verification. I assume that this has been done as part of some sort of testing infrastructure, so should be trivial to add to the paper*

Response: We would incorporate a section in discussion on model verification (Section 4.3)

**Section 4.3.  Model evaluation in West Falmouth Harbor**

20   In order to qualitatively evaluate the seagrass growth model, we have compared the modeled results with observations by del Barrio et al. (2014) that measured the extent of seagrass coverage in West Falmouth Harbor (red outline in Fig. 11). The field data is only available for the northern region of WFH where the model-data comparisons are performed. The model results are compared by extracting the peak above ground biomass (AGB) on 14th day of the simulation

25   and normalized with the initial above ground biomass. The ratio of $AGB/AGB_{initial}$ is considered as a representative of seagrass growth. We assume that for $AGB/AGB_{initial} > 1$, there is a potential for seagrass growth and for $AGB/AGB_{initial} < 1$, the conditions are unfavorable for seagrass growth. In fig 11, the model and field data show a 89% agreement to determine the seagrass growth or dieback. The western region of outer harbor shows seagrass growth potential

30   and agrees with the extent that the seagrass coverage is observed. In the eastern region, the field

data shows no seagrass coverage and the model also predicts potential seagrass dieback. The model predicts seagrass dieback because of nitrate loading from shoreline point sources that leads to increased chlorophyll and light attenuation (figures 8a, b). The model and observations do not compare well in the central basin of outer harbor where the model shows seagrass dieback potential while the field data shows presence of seagrass. In the central basin, the field data shows the presence of seagrass while its density remains low in this region. On the other hand, the modelled seagrass suffers dieback due to the bathymetric controls in the deeper central basin (decreased near-bottom PAR Fig. 8c).

Direct estimates of above ground SAV biomass have also been recently made in West Falmouth Harbor (Hayn et al., unpublished data). Although these measurements were not made during the same year as our simulations (measurements in 2006, 2007, 2013; model 2010), the mean above ground biomass measured in the outer harbor of 49.5 (June 21-July 6 2006), 45.3 (June 6-19 2007), and 41.5 g C m$^{-2}$ (July 15-19 2013) is consistent with the range of model simulations during a comparable period (July 2-19) in the outer (28.1 to 51.1 g C m$^{-2}$) and middle (14.9 to 37.4 g C m$^{-2}$) harbors. The July 2-19 model range of 45.7 to156.3 mmol N m$^{-2}$ across the middle and outer harbor is also consistent with annual mean *Z. marina* biomass (10-88 mmol N m$^{-2}$) reported in nearby shallow systems on Cape Cod (Hauxwell et al. 2003) assuming a literature-based average that above ground SAV biomass is 1.5% N. The range in the model is computed based on the minimum and maximum values of AGB during the 18 day simulation period.

[Figure]

Fig 11: Modeled $AGB/AGB_{initial}$ (above ground biomass) distribution compared with field data showing seagrass coverage extent (red solid line). Values of $AGB/AGB_{initial} > 1$ represent seagrass growth potential and below 1 indicate potential seagrass decline at day 14 of the simulation.

*Comment 2: Check code availability and make it clearer which parts of COAWST are a part of the paper. As the editor has indicated, a Zenodo archive, coupled with some indication of which code this paper refers to would be a great help*

Response:

We have followed the official USGS policy to archive and release the model. These links detail the process of going through a review and approval process to release USGS software:

https://www.usgs.gov/about/organization/science-support/survey-manual/im-osqi-2016-01-review-and-approval-software

https://github.com/usgs/best-practices

Following these policy steps, the source code was made available for distribution at https://code.usgs.gov/coawstmodel/COAWST.

The major code development that was done for this project is contained within the COAWST folder on the following path.

"https://code.usgs.gov/coawstmodel/COAWST/blob/master/ROMS/Nonlinear/Biology/"

This folder contains several methods of computing water column biogeochemistry. Other than the I/O component of our implementation, the algorithmic development in this study only modifies two files on this path: "estuarybgc.h" and "sav_biomass.h". The file "sav_biomass.h" contains all the newly added equations for the growth of SAV based on the nutrient loading in the water column. The forcings to the SAV growth model (temperature, light, nutrient availability, exchanges nutrients, detritus, dissolved inorganic carbon, and dissolved oxygen) are provided through the file "estuarybgc.h" that calls "sav_biomass.h". The file "estuarybgc.h" solves for the water column biogeochemistry and was based on existing modelling framework developed by Fennel et al. (2006) (also coded as "fennel.h").

Other important paths that existed in the framework prior to the current modeling effort but are being used in the modeling process include:

1. "https://code.usgs.gov/coawstmodel/COAWST/blob/master/ROMS/Nonlinear"-
The main kernel of the 3-D non-linear Navier-Stokes equations is contained within this part and links all the submodels: biological, vegetation and sediment models.

2. "https://code.usgs.gov/coawstmodel/COAWST/blob/master/ROMS/Nonlinear/Vegetation/"
The kernals that account for seagrass-hydrodynamics interactions.

3. "https://code.usgs.gov/coawstmodel/COAWST/blob/master/ROMS/Nonlinear/Sediment/"
The kernals that account for sediment transport.

This information is also added in the code availability section of the current manuscript.

*Comment 3: Equations - Equations in 2.2 are very difficult to read with "words" being used as symbols in a lot of cases; especially when "lim" is used in a symbol it makes it difficult to know of this is the mathematical limit of or a symbol at a glance. Symbols such as lambda_SAVmax (eq 3) should be altered to remove operation symbols from them. There are also symbols such as kl. Is this k * l or a symbol kl? I would recommend the use of single symbols where possible and remove as many "words" as possible. Same applies to table 1.*

Response:

"lim" is a symbol in the equations and is not defining a mathematical limit. To avoid confusion, it has been replaced with the symbol "lmt".

"$\lambda_{SAV-max}$: Removed the dashed part in the symbol name and the new one is $\lambda_{SAV,max}$. We did the same change to other variables that had the same issue such as $\lambda_{EPB,max}$

"kl" – This symbol is changed $klmt$ i.e. the half-saturation for light limitation. The "$lmt$" part is then consistent with the symbol of light limitation.

The reason we used multiple letters in the equations is to be consistent with the legibility of the code. In the larger framework of the COAWST model where there are several variables, single letter symbols do not suffice.

*Minor comments:*

*Comment 1: Recent observational studies have addressed feedbacks between SAV meadows, current velocity, sedimentation, and nutrient cycling and suggest SAV are ecosystem engineers whose growth can be self-reinforcing.*

Response: Modified to : "Recent observational studies have addressed feedbacks between SAV meadows and their role in modifying current velocity, sedimentation, and nutrient cycling."

*Comment 2: Modelled SAV biomass is represented as a function of temperature, light, and nutrient availability and exchanges nutrients, detritus, dissolved inorganic carbon, and dissolved oxygen with the water-column biogeochemistry model.*

Response: This sentence is split into two sentences.

"Modelled SAV biomass is represented as a function of temperature, light, and nutrient availability. The modelled SAV community exchanges nutrients, detritus, dissolved inorganic carbon, and dissolved oxygen with the water-column biogeochemistry model."

*Comment 3: Line 25, pg 2 – extra( ) round reference*

Response: The lines 22-25 were altered to remove the extra () reference.

These processes were implemented within the open source COAWST (Coupled-Ocean-Atmospheric-Wave-Sediment Transport) modelling system (Warner et al., 2010) that couples the hydrodynamic model (ROMS), the wave model (SWAN) and the Community Sediment Transport Modelling System (CSTMS).

Response: Could not find this typo.

5    *Comment 5: Figure 3 Remove orientation axis. Its plan view, so z isn't on !*

Response: Removed the axis

[Figure]

*Comment 6: Figure 4 – Capital letters in axes title*

Response: Fixed this in the figure.

[Figure]

(a)

[Figure]

(b)

**Figure 4: Planform view of (a) depth-integrated SSC, (b) light attenuation averaged over the last day of the simulation in the idealized domain.**

*Comment 7: Figure 5 – Triangle and dot not explained in caption. Capital letters in axes title*

Response:  Red dot and blue star represent two points that are located at 0.1 km and 4.5 km into the SAV bed.

[Figure]

(a)

(b)

**Figure 5: Planform view of (a) above ground biomass and (b) vegetation stem density averaged over the last day of the simulation in the idealized domain. Red dot and blue triangle represent two points that are located at 0.1 km and 4.5 km into the SAV bed respectively.**

*Comment 8: Figure 6- Capital letters in axes titles. Remove "Figure" from subcaptions*
Response:

[Figure]

(a)                                                    (b)

(c)                                                    (d)

**Figure 6: Time-series of a) light attenuation, b) above ground biomass, c) net primary production of SAV ($pp_{SAV} - agar_{SAV} - bgr_{SAV}$), and d) SSC in the bottom cell averaged every day from the two locations identified in Fig. 5a.**

*Comment 9: Figure 7 – Capital letters in axes titles*

Response:

[Figure]

(a)

(b)

**Figure 7. Magnitude of bottom stress (left) and depth-integrated SSC (right) at the end of the simulation plotted along the y axis of the idealized domain at two locations, including one outside (x=1.8 km; panel a) and one inside the SAV bed (x=4.8 km, panel b)**.

*Comment 10: Figure 8 – replace color scheme with color-blind friendly scheme.*

Response: Used the "balance" map from the cmocean package

[Figure]

**(a)**  **(b)**

**(c)**  **(d)**

**Figure 8. Mean over 22 days of a) depth-averaged chlorophyll, b) light attenuation, c) near-bottom PAR, and d) peak above ground biomass at day 14 of the simulation. Red circle indicated outer harbor (left) and blue triangle indicated inner harbor (right) points for time-series data in Figure**

*Comment 11: Figure 10 – as above*

Response: Used the "balance" map from the cmocean package

[Figure]

**(a)**

**(b)**

**(c)**

**(d)**

**Figure 10. Change in outcomes between impacted and non-impacted scenario (nitrate loading scenario – no loading scenario). Difference in mean over 22 days of (a) depth-averaged chlorophyll, (b) light attenuation, (c) near-bottom PAR, and (d) peak above ground biomass at day 14 of the simulation**.

**2.2 Author response to Referee 2**

We thank you the reviewer for their suggestions. The response to the reviewers comments are in black while the original comments are in *blue*.

*1. Mathematical notation*

*1.1 Mathematical symbol names should be single letters (Latin, Greek, or potentially from another alphabet if really needed). Using multi-letter names creates confusion about what is a variable name and what is a multiplication of symbols. This is a convention that very much also holds in the marine biogeochemistry modelling community, for example the NPZD model is named after the conventional (single letter) symbol names for its four prognostic quantities).*

Response: The reason we used multiple letters in the equations is to be consistent with the legibility of the code. In the larger framework of the COAWST model where there are several variables, single letter symbols do not suffice.

*1.2. If it is necessary or useful to use a multi-letter subscript or superscript to further identify a variable, then this should be typeset in upright letters to avoid the confusion with a product of symbols. Using LATEX, this can be achieved with \mathrm, for example Topt is written as $T_{\mathrm{opt}}$.*

35 Response: We have replaced all the subscripts and superscripts with upright letters. Please see a revised version of Section 2.2 at the end of this response.

*1.3. exp is the exponential function, it takes its argument in round brackets and not as an index. e is a number, the base of natural logarithms, and can be exponentiated by writing an index. The current mix of these two notations, for example in equation 2, is at best confusing and at worst meaningless*

Response: We have used exp as a function in the equations now with its arguments in brackets.

*1.4 . Mathematical function names are typeset upright and usually use lower case letters, for example exp, min ($\exp$, and $\min$ respectively).*

Response: We have used lower case letters for all the mathematical functional names (please see revised section 2.2).

*1.5 Double subscripts should be avoided where possible. If they are unavoidable then they should not be separated by a hyphen, because a horizontal line universally means subtraction. A comma, possibly augmented by brackets of some type, would be a better choice.*

Response: We have eliminated hyphen as per the review and used a comma to describe the double subscripts.

*2. Equations and discretisation*
*The introduction to section 2.2 claims that the remainder of the section will introduce the equations solved. In fact, we are only treated to a disconnected set of source terms for an unspecified set of equations. Please provide the full set of differential equations being solved, before going into detail about the definition of the terms. In addition, the equations are clearly being solved numerically, so a complete model description also requires the inclusion of the discretisation used, and how the resulting discrete linear or nonlinear system is solved.*

Response: We would add a modified section 2.3 for connecting the SAV growth model that provides the source terms to the complete model description. It mentions the integration of

source terms into the water-column biogeochemistry model and the discretization methods of the resulting system of equations.

*3. Verification and evaluation There is effectively no verification or validation of the model. The test cases provided are purely descriptive: the model is run and the authors describe what happened. This does not provide suitable evidence either that the model is correctly implemented, or that it is realistic. The usual way of demonstrating the former would be using the method of manufactured solutions (MMS) to create artificial analytical solutions to the system, and then demonstrating convergence to them at the expected rate. For more information on MMS see Farrell et al. (2011) section 4.1 (https://doi.org/10.5194/gmd-4-435-2011).*
*In order to provide some level of evaluation of the model, it would be necessary to present a qualitative or quantitative comparison of the model to an external reference. The external reference might be directly with observational data, or might be with the results of another well-evaluated model. In any event, an external comparator is absolutely necessary.*

Response:

3.1 Verification

The process of manufactured solutions to create artificial analytical solutions is possible where an analytical solution of a physical problem is available and convergence of the solution to the partial differential equation can be tested. The authors acknowledge that similar verification ideas are the way to validate test cases. In the current work, we used modified an existing point model (Madden and Kemp, 1996) that calculated changes in vegetation biomass that we have adapted to predict changes in vegetation properties (density and height) that impact physical processes in the model (e.g., advection, resuspension). The point model was implemented with the inclusion of spatial variation in the 3-D model. There is no analytical solution to the point model that we developed and we can only verify the implementation of a point model in the 3-D framework by running the point model separately and running the 3-D model after turning off the hydrodynamics, sediment dynamics along with the advection-diffusion processes (i.e. stripping the 3-D model down to be a point model). Alternatively, the idealized domain can be utilized within the 3-D model to show the sensitivity of using individual components of the model for eg. turning the sediment model off to show that a better light climate can provide

better environment for SAV to grow. The overarching goal of the idealized case in the manuscript is to demonstrate that the model is capable of simulating expected dynamics that included process of seagrass growth and dieback, its effect on sediment and hydrodynamics processes (i.e. two way feedback between the hydrodynamics-sediment-biological) dynamics. However, in lieu of this type of evaluation of the model, we are providing a comparison of modeled vegetation properties with independently-collected field data for the case of West Falmouth Harbor in the revised manuscript. In order to do build this qualitative (SAV distribution) and quantitative (SAV biomass at specific locations) comparison with external data, we will incorporate a new section (Section 4.3) to perform the model evaluation. Typically, the coupled biological-sediment models are assessed in a similar manner (Matsumoto et al., 2013, Cossarini et al. 2017, Sherwood et al., 2018).

**3.1 Marked up manuscript that tracks changes in MS-Word Format including the changes:**

[revised manuscript text omitted]

**Commented [KT(S2):** Adjusted round references as per reviewer 1.

**Commented [KT(S3):** Changes in paragraph 2 as per reviewer 1 suggestions.

ROMS, the presence of SAV extracts momentum, adds wave-induced streaming, and generates turbulence dissipation. Similarly, the wave dissipation due to vegetation modifies the source term of the action balance equation in SWAN. All these sub-grid scale parameterizations account for changes due to vegetation in the water column extending from the bottom layer to the height of the vegetation. SWAN only accounts for wave dissipation due to

5   vegetation at the bottom layer. The coupling between the two models occurs with an exchange of water level and depth averaged velocities from ROMS to SWAN and wave fields from SWAN to ROMS after a desired number of time steps. The vegetation properties are separately input in the two models at the beginning of the simulations. Through these changes, the SAV can affect the bottom stress calculations that determine the resuspension and transport of sediment, providing a feedback loop between SAV-sediment dynamics-hydrodynamics and wave

10  dynamics. To account for sediment dynamics, the Community Sediment Transport Modelling System (CSTMS) (Warner et al., 2010) is used to track the transport of suspended-sediment and bed load transport under the action of current and wave-current forcing. The model can represent an unlimited number of user defined sediment classes.

**2.2   SAV growth model**

15  The SAV growth model is primarily based upon a previous growth model developed and implemented in Chesapeake Bay by Madden and Kemp (1996). The model simulates the temporal dynamics of above ground biomass (AGB) that consists of stems or shoots, and the below ground biomass (BGB) that consists of roots or rhizomes. In addition to AGB and BGB, epiphytic algal biomass (EPB) is simulated to account for reductions in light availability to plant leaves due to shading of SAV leaves by epiphytes under high nutrient loading conditions. AGB, BGB and EPB are

20  simulated as total biomass per unit area, with nitrogen as the currency for biomass. Changes in AGB and BGB pools are simulated as a function of primary production and respiration, mortality (e.g., grazing), and nitrogen exchange through the seasonal translocation of nitrogen between roots and shoots. EPB are modelled as a function of primary production, respiration, and mortality.

The remaining section describes the source terms that calculate the evolution of AGB, BGB and EPB. The default
25  input parameters required by the following model equations are described in Table 1.

30  ~~Kemp (1996). The model simulates the temporal dynamics of above ground biomass (AGB) that consists of stems or shoots, and the below ground biomass (BGB) that consists of roots or rhizomes. In addition to AGB and BGB, epiphytic algal biomass (EPB) is simulated to account for reductions in light availability to plant leaves due to shading of SAV leaves by epiphytes under high nutrient loading conditions. AGB, BGB and EPB are simulated as total biomass per unit area, with nitrogen as the currency for biomass. Changes in AGB and BGB pools are~~
35

**Commented [KT(S4):** Made changes to all the equations as per reviewer 1 and 2 suggested. The combined changes include:

For reviewer 1:
1. lim" is a symbol in the equations and is not defining a mathematical limit.
To avoid confusion, it has been replaced with the symbol "lmt".
"$\lambda_{(SAV-max)}$: Removed the dashed part in the symbol name and the new one is $\lambda_{(SAV,max)}$.

2. Removed the dashed part in the symbol name and the new one is $\lambda_{(SAV,max)}$. We did the same change to other variables that had the same issue such as $\lambda_{(EPB,max)}$

3. "kl" – This symbol is changed klmt i.e. the half-saturation for light limitation. The "lmt" part is then consistent with the symbol of light limitation.

For reviewer 2:

1. used exp as a function in the equations now with its arguments in brackets.

2.replaced all the subscripts and superscripts with upright letters

3. eliminated hyphen as per the review and used a comma to describe the double subscripts

~~through the seasonal translocation of nitrogen between roots and shoots. EPB are modelled as a function of primary production, respiration, and mortality. The remaining section describes the model equations used to simulate AGB, BGB and EPB and how they were implemented within a three-dimensional framework. The default input parameters required by the following model equations are described in Table 1.~~

2.1 Primary production ($pp_\text{SAV}$): The primary production of AGB depends on the maximum potential growth rate ($ua$) and downward deviations from this maximal rate resulting from light ($llmt_\text{SAV}$) and nutrient ($nlmt_\text{SAV}$) availability as:

$$pp_\text{SAV} = ua \, \min(lmt_\text{SAV}, nlmt_\text{SAV}) \tag{1}$$

The maximum potential growth ($ua$) can be described as:

$$ua = \lambda_\text{SAV} \, nlmt_\text{SAV} \, scl \, \exp\left[arc\left(\frac{1.0}{T - T_\text{opt}}\right)\right] \tag{2}$$

[revised manuscript text omitted]

Commented [KT(S10):] Capital letter in axis title.

[Figure]

(a)

(b)

(c)

(d)

[Figure]

Figure (a)                                        Figure (b)

[Figure]

Figure (c)                                        Figure (d)

**Figure 6: Time-series of a) light attenuation,b) above ground biomass, c) net primary production of SAV**

5  $(pp_{SAV} - agar_{SAV} - bgr_{SAV})$**, and d) SSC in the bottom cell averaged every day from the two locations identified in Fig. 5a.**

[Figure]

Figure (a)

[Figure]

Figure (b)

**Figure 7. Magnitude of bottom stress (left) and depth-integrated SSC (right) at the end of the simulation plotted along the y axis of the idealized domain at two locations, including one outside (x=1.8 km; panel a) and one inside the SAV bed (x=4.8 km, panel b)**.

Commented [KT(S11):] Using the cmocean package as per reviewer 1 comments

(a)                                                              (b)

[Figure]

(c)                                                              (d)

[Figure]

**Figure 8. Mean over 22 days of a) depth-averaged chlorophyll, b) light attenuation, c) near-bottom PAR, and d) peak above ground biomass at day 14 of the simulation. Red circle indicated outer harbor (left) and blue triangle indicated inner harbor (right) points for time-series data in Figure 9.**

[Figure]

**Figure 9. Time-series of a) chlorophyll, b) light attenuation, c) near-bottom PAR, and d) above ground biomass from outer and inner harbor locations identified in Figure 6.**

[Figure]

**(a)** **(b)**

[Figure]

**(c)** **(d)**

Figure 10. Change in outcomes between impacted and non-impacted scenario (nitrate loading scenario – no loading scenario). Difference in mean over 22 days of (a) depth-averaged chlorophyll, (b) light attenuation, (c) near-bottom PAR, and (d) peak above ground biomass at day 14 of the simulation.

[Figure]

**Commented [KT(S12):** Added this new figure corresponding to section 4.3 for model evaluation as per reviewer 1 and 2 points.

[revised manuscript text omitted]

---

## Author Response (AR2)

Dear Editor,

I am pleased to resubmit the manuscript "Development of a Submerged Aquatic Vegetation
Growth Model in a Coupled Wave-Current-Sediment-Transport Modeling System (COAWST
v3.4)" to be considered for publication in Geophysical Model Development.

We have incorporated the minor revisions in this version. There were two issues presented by the
reviewers after the last submission.
1. Releasing the model version of the source code on a platform such as sciencebase
2. Revising the equations to avoid multi-letter symbols

The authors with the help of other USGS staff have created a copy of the model source code that
was used in this work and uploaded it on an independent sciencebase link following the editor's
suggestions. We have avoided multi-letter symbols in all equations and added a table in the paper
that explains the equivalent symbol in the source code. The sub-sections in the document address
the suggestions from both the referee 1 and the editor.

1.1 Authors response to the editor, Pages 1-3
1.2 Authors response to referee, Pages 4-5
1.3 Marked up manuscript with changes based on reviewers comments
1.4 Final manuscript with all the changes included

Sincerely,
Tarandeep

**1.1. Authors response to the editor**

We have created a sciencebase link to the current source code as a tarball. We have the code
availability section as follows:

Changes in the paper:

**6 Code availability**

The implementation of the SAV growth model has been implemented in the Coupled Ocean Atmosphere Waves Sediment-Transport Modeling System (COAWST v3.4). This particular version is available for download at: https://www.sciencebase.gov/catalog/item/5f15d69082cef313ed81996a. Users are encouraged to download COAWST distributed through the USGS code archival repository. It is available for download on https://code.usgs.gov/coawstmodel/COAWST. The COAWST distribution files contain source code derived from ROMS, SWAN, WRF, MCT and SCRIP, along with the Matlab code, examples and a User's Manual.

The major code development that was done for this project is contained within the COAWST folder on the following path.

"https://code.usgs.gov/coawstmodel/COAWST/blob/master/ROMS/Nonlinear/Biology/"

This folder contains several methods of computing water column biogeochemistry. Other than the I/O component of our implementation, the algorithmic development in this study only modifies two files on this path: "estuarybgc.h" and "sav_biomass.h". The file "sav_biomass.h" contains all the newly added equations for the growth of SAV based on the nutrient loading in the water column. The forcings to the SAV growth model (temperature, light, nutrient availability, exchanges nutrients, detritus, dissolved inorganic carbon, and dissolved oxygen) are provided through the file "estuarybgc.h" that calls "sav_biomass.h". The file "estuarybgc.h" solves for the water column biogeochemistry and was based on existing modelling framework developed by Fennel et al. (2006) (also coded as "fennel.h").

Other important paths that existed in the framework prior to the current modeling effort but are being used in the modeling process include:

1. "https://code.usgs.gov/coawstmodel/COAWST/blob/master/ROMS/Nonlinear"-

The main kernel of the 3-D non-linear Navier-Stokes equations is contained within this part and links all the submodels: biological, vegetation and sediment models.

2. "https://code.usgs.gov/coawstmodel/COAWST/blob/master/ROMS/Nonlinear/Vegetation/"

The kernals that account for seagrass-hydrodynamics interactions.

5  3. "https://code.usgs.gov/coawstmodel/COAWST/blob/master/ROMS/Nonlinear/Sediment/"

The kernals that account for sediment transport.

**1.2 Authors response to referee 1**

The response to the Reviewer's comments are in black while the original comments are in blue.

Minor revisions:

Overall - please avoid using mathematical notation in the form of multiple letters. This happens throughout the paper. For example equation 3 has the term AGB. I can't tell if that is A*G*B or a single term (it's the latter, but this is only clear when reading the full paper). Please use a single symbol, e.g. phi to denote terms like this. All equations are affected by this, but it is only a minor change throughout.

Authors response: We have taken out the multi-letter symbols throughout the manuscript and added a new table 1 and modified table 2 to let the readers know about the equivalent symbols in the source code for the model variables and model input parameters respectively. An example of this is shown with an excerpt from the manuscript:

" The remaining section describes the source terms that calculate the evolution of AGB, BGB and EPB and denoted by $\alpha$, $\beta$ and $\gamma$ respectively. Table 1 and Table 2 describe model variables and input parameters along with their equivalent symbols used in the source code.

2.1 Primary production ($\rho_s$): The primary production of $\alpha$ depends on the maximum potential growth rate ($\mu_s$) and downward deviations from this maximal rate resulting from light ($\varphi_s$) and nutrient ($\vartheta_s$) availability as:

$$\rho_s = \mu_s \min(\varphi_s, \vartheta_s)"$$

Pg 1, line 15. Additional period after "cycling. . To..."

Authors response: Fixed this space as

"and nutrient cycling. To represent these dynamic processes in a numerical model, the presence of SAV and its effect on hydrodynamics (currents and waves) and sediment dynamics was incorporated into the open source model COAWST. "

pg 2, line 8 - check the in text citation formatting. Not sure this is correct. Check throughout.

Authors response: We replaced ";" by "and"

Authors response: Fixed this as:

5   Changes in AGB and BGB pools are simulated as a function of primary production and respiration, mortality (e.g. grazing), and nitrogen exchange through the seasonal translocation of nitrogen between roots and shoots. EPB are modelled as a function of primary production, respiration, and mortality.

10   Authors response: Fixed this.

pg 5 line 17. Does there need to be a new paragraph for this?

Authors response: Fixed this issue at all places

Authors response: Same as above

pg 8: lines 12-15: Some repetition with the intro so this can be trimmed down a little.

Authors response: We have modified these lines

Authors response: Changed this to have $NO_3$ (Nitrate), $NH_4$ (Ammonium) in the paper

25   Authors response: Fixed this.

1.3 Marked up manuscript

[revised manuscript text omitted]

---

## Author Response (AR3)

Dear Editor,

We have incorporated the minor revisions in this version by removing the weblinks, citing the data
releases, and adding them to the bibliography section.

Sincerely,
Tarandeep

1.1 Marked up manuscript

[revised manuscript text omitted]